# Efficient oxygen evolution electrocatalysis in acid by a perovskite with face-sharing $IrO_6$ octahedral dimers

Lan Yang[1,2], Guangtao Yu[3], Xuan Ai [1], Wensheng Yan [4], Hengli Duan [4], Wei Chen[3], Xiaotian Li[2], Ting Wang[3], Chenghui Zhang[3], Xuri Huang[3], Jie-Sheng Chen[5] & Xiaoxin Zou [1]

The widespread use of proton exchange membrane water electrolysis requires the development of more efficient electrocatalysts containing reduced amounts of expensive iridium for the oxygen evolution reaction (OER). Here we present the identification of 6H-phase $SrIrO_3$ perovskite (6H-$SrIrO_3$) as a highly active electrocatalyst with good structural and catalytic stability for OER in acid. 6H-$SrIrO_3$ contains 27.1 wt% less iridium than $IrO_2$, but its iridium mass activity is about 7 times higher than $IrO_2$, a benchmark electrocatalyst for the acidic OER. 6H-$SrIrO_3$ is the most active catalytic material for OER among the iridium-based oxides reported recently, based on its highest iridium mass activity. Theoretical calculations indicate that the existence of face-sharing octahedral dimers is mainly responsible for the superior activity of 6H-$SrIrO_3$ thanks to the weakened surface Ir-O binding that facilitates the potential-determining step involved in the OER (i.e., $O^* + H_2O \rightarrow HOO^* + H^+ + e\text{-}$).

[1] State Key Laboratory of Inorganic Synthesis and Preparative Chemistry, College of Chemistry, Jilin University, 130012 Changchun, People's Republic of China. [2] College of Materials Science and Engineering, Jilin University, 130022 Changchun, People's Republic of China. [3] Laboratory of Theoretical and Computational Chemistry, Institute of Theoretical Chemistry, Jilin University, 130023 Changchun, People's Republic of China. [4] National Synchrotron Radiation Laboratory, University of Science and Technology of China, 230029 Hefei, Anhui, People's Republic of China. [5] School of Chemistry and Chemical Engineering, Shanghai Jiao Tong University, 200240 Shanghai, People's Republic of China. These authors contributed equally: Lan Yang, Guangtao Yu. Correspondence and requests for materials should be addressed to W.C. (email: w_chen@jlu.edu.cn) or to X.Z. (email: xxzou@jlu.edu.cn)

The oxygen evolution reaction (OER) is the primary reaction that occurs at the anode in many electrochemical energy conversion processes, such as water splitting, $CO_2$ reduction, $N_2$ fixation, etc[1–3]. Due to its multi-proton/electron-coupled kinetics, the OER is a quite sluggish half-reaction, and often has a crucial role in the overall efficiency of these electrochemical processes[1,4,5]. Therefore, over the past several years, considerable effort has been devoted to the search and synthesis of efficient electrocatalysts that can significantly lower the kinetic barriers for OER. Whereas a large number of oxygen evolution electrocatalysts based on a wide range of transition metals work well under alkaline conditions[6–13], only the few of the recently developed electrocatalysts are found to be effective for OER in acidic media[14–23]. Despite the difficulties, there are strong demands for the OER in acidic pH regime. For instance, proton exchange membrane (PEM) water electrolysis is an attractive and advanced route for sustainable hydrogen production[24], especially when coupled with some renewable energy systems. PEM water electrolysis offers many advantages, such as high current densities, ultrahigh gas purities, low ohmic losses and good compactness[24]. But this technique requires corrosion-resistant electrocatalysts that can operate efficiently in strongly acidic media.

The reasons for the slow development of OER electrocatalysts operating in acid can be reflected by the facts that: first, the great majority of promising electrocatalysts, including those can work well in alkaline media, are chemically unstable in acid[6–13]; second, the corrosion/decomposition of electrocatalysts in acid always get worse under a strongly oxidative electrocatalysis condition; third, even for the few with relatively good catalytic stability in acid, uncontrolled surface reconstruction often take place during electrocatalysis, so that the accurate elucidation of the atomic basis for catalytic properties is very difficult[16,17]. At present, iridium-based oxides, especially $IrO_2$, are generally considered the only electrocatalysts with reasonable activity and stability for the acidic OER. So, considering the high cost and low

Earth abundance of iridium, researchers are in pursuit of corrosion-resistant, structurally stable, more active oxygen evolution electrocatalysts containing reduced iridium amount.

Herein we identify 6H-SrIrO₃, containing 27.1 wt% less iridium than $IrO_2$, as a highly efficient electrocatalyst with excellent structural stability for OER in acid. The material gives the lowest overpotential at 10 mA $cm_{geo}^{-2}$ (current densities per geometric area), and exhibits the highest iridium mass activity for the acidic OER among the iridium-based oxide catalysts reported recently. To the best of our knowledge, an iridium-based compound (or an unusual perovskite oxide) with face-sharing $IrO_6$ octahedral subunits has never been reported to effectively electrocatalyze OER in acid. In addition, 6H-SrIrO₃ has never been studied as a catalyst in any chemical reactions.

## Results

**Crystal structure and electrical conductivity of 6H-SrIrO₃.** 6H-SrIrO₃ is the thermodynamically stable polymorph of SrIrO₃, and adopts a monoclinic distortion structure of the hexagonal form of perovskite BaTiO₃ (Fig. 1a)[25]. It consists of alternating face-sharing $IrO_6$ octahedral dimers and corner-sharing, isolated $IrO_6$ octahedra along the $c$ axis (Fig. 1b). Correspondingly, there are two types of Ir atoms in 6H-SrIrO₃. One is in the face-sharing $IrO_6$ octahedral dimers, and the other is in the corner-sharing, isolated $IrO_6$ octahedra (Fig. 1c). It is remarkable that there is Ir–Ir metallic bonding in face-sharing $IrO_6$ octahedra, as revealed by the very short Ir(I)–Ir(I) distance (2.796 Å). This can be further supported by the electron location function (ELF)[26,27]. As presented in Fig. 1d, the correlative ELF values with a range of 0.4 to 0.5 distribute between the two neighboring Ir atoms in face-sharing $IrO_6$ dimers, indicating that Ir–Ir bond is metallic in character. (See details about ELF in Methods). Besides Ir–Ir metallic bonding, there are obviously weaker Ir–O bonds in the face-sharing $IrO_6$ octahedral dimers in comparison with those in the corner-sharing, isolated $IrO_6$ octahedra. As shown in Fig. 1c

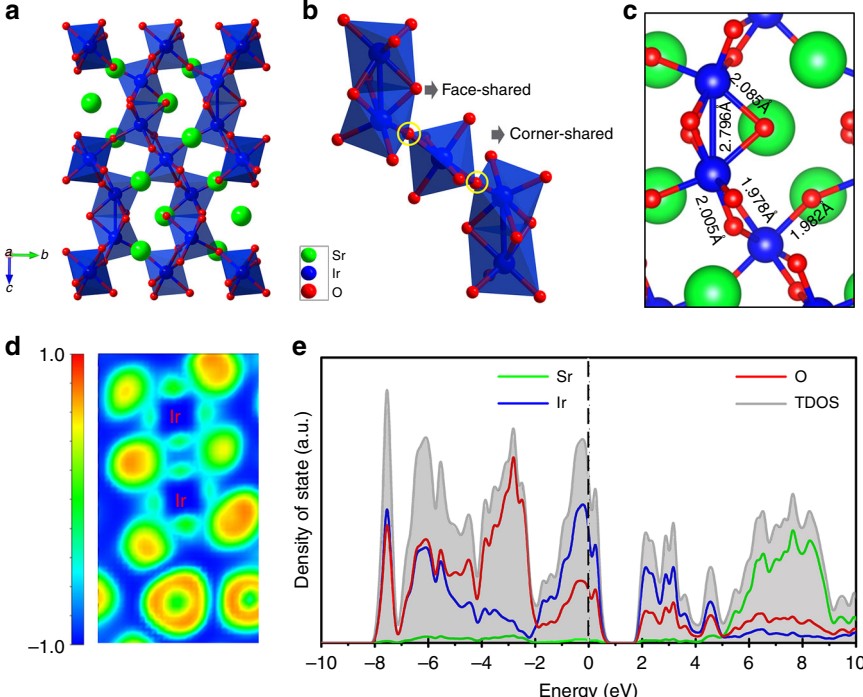

**Fig. 1** Crystal structure and electronic structure of 6H-SrIrO₃. **a** Crystal structure of 6H-SrIrO₃. **b** A local connection pattern of $IrO_6$ octahedra, in which face-sharing $IrO_6$ octahedral dimers and corner-sharing, isolated $IrO_6$ octahedron are shown. **c** Local ball-and-stick model of 6H-SrIrO₃, in which typical Ir–Ir and Ir–O bond lengths are presented. **d** Plot of electron location function for Ir–Ir bonding in 6H-SrIrO₃. **e** Density of states of 6H-SrIrO₃, in which the Fermi level is 0 eV

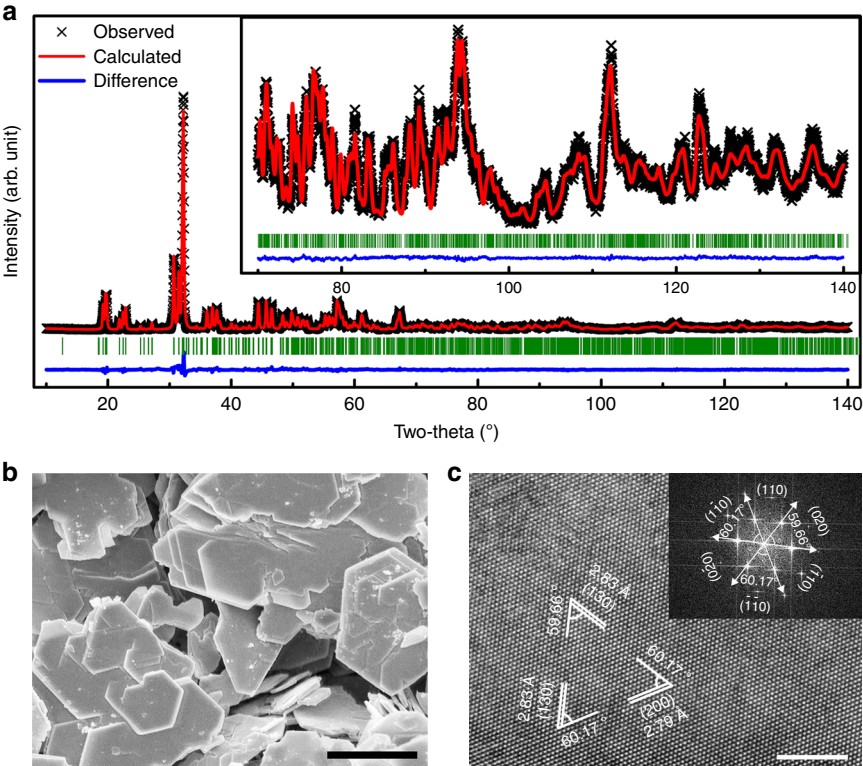

**Fig. 2** Structural characterizations of 6H-SrIrO₃. **a**, XRD pattern with a refinement plot of 6H-SrIrO₃. **b**, SEM image of 6H-SrIrO₃. Scale bar, 2 µm. **c**, HRTEM image of 6H-SrIrO₃. Scale bar, 5 nm. Inset: the corresponding fast Fourier transform image

and Supplementary Figure 1, the Ir–O bond lengths in the former, ranging from 2.005 to 2.085 Å, are much larger than those in the latter (1.978–1.982 Å, close to those in IrO₂). The face-sharing IrO₆ octahedra and the unique bonding features in them, including the Ir–Ir metallic bonding and the weak Ir–O bonding, mark 6H-SrIrO₃ off from other Ir-based oxide catalysts reported previously (e.g., IrO₂, see its crystal structure in Supplementary Figure 2)[15–22].

The computed density of state (DOS) for 6H-SrIrO₃ reveals that this material has intrinsic metallic conductivity[25,28], which is reflected by the density of state from Ir and O atoms crossing the Fermi level (Fig. 1e). This is further supported by the experimental result (Supplementary Figure 3). The electrical resistivity of 6H-SrIrO₃ was measured to be ~2.7 × 10⁻⁴ Ω m at room temperature. The metallic conductivity of 6H-SrIrO₃ is believed to be beneficial for electrochemical applications.

**Synthesis and electrochemical properties for OER in acid of 6H-SrIrO₃.** Encouraged by the unique crystal structure of 6H-SrIrO₃, we synthesized this material and attempted to explore its potential electrocatalytic activity for OER in acid. The observed powder X-ray diffraction (XRD) pattern is almost in agreement with the fitting plot (Fig. 2a), revealing that the material is composed of high purity 6H-SrIrO₃. The refinement results are listed in Supplementary Table 1. Scanning electron microscopy (SEM) images (Fig. 2b and Supplementary Figure 4) show that the material has micron-sized, plate-like particles with a dominant thickness distribution of 30–60 nm. Additionally, chemical mapping and energy dispersive X-ray spectroscopy (EDS) show that the Sr and Ir elements are homogeneously distributed over the material, with a Sr:Ir atomic ratio of ca. 1:1 (Supplementary Figure 5). X-ray photoelectron spectroscopy (XPS) result further reveals a Sr:Ir atomic ratio of ca. 1:1 for 6H-SrIrO₃. In the high resolution transmission electron microscopy (HRTEM) images

(Fig. 2c and Supplementary Figure 6), three sets of lattice fringes are shown, giving interplanar distances of 0.283, 0.279, and 0.283 nm corresponding to the (130), (200), and (1̄30) crystallographic planes of 6H-SrIrO₃, respectively. The observed angles between these crystallographic planes match well with the theoretical values. These results further suggest that the exposed facet of micron-sized 6H-SrIrO₃ plates is {001}. The exposure of this facet is also confirmed by the fast Fourier transform image (Fig. 2c, inset). Furthermore, high-angle annular dark field (HAADF) STEM image (Supplementary Figure 7) of the edge of a 6H-SrIrO₃ particle again reveals that the 6H-SrIrO₃ is highly crystalline.

Before the electrocatalytic studies, we first tested the chemical stability of 6H-SrIrO₃ in strongly acidic media (0.5 M H₂SO₄). XRD and TEM results reveal that 6H-SrIrO₃ retains its crystal structure well, even after its exposure in 0.5 M H₂SO₄ over 48 days (Supplementary Figure 8). In addition, after such a long time testing, there is no detectable leached Ir species and only ~1.3% of total Sr content leached in the acidic solution. These results suggest that 6H-SrIrO₃ has excellent chemical stability in acid.

Next, we evaluated the electrocatalytic properties of 6H-SrIrO₃ toward OER in acidic media (0.5 M H₂SO₄). For comparative purpose, we also synthesized IrO₂ nanoparticles with a particle size of 10–20 nm (see structural characterizations in Supplementary Figure 9) and studied their catalytic activity for OER. As shown in Fig. 3a, both 6H-SrIrO₃ and IrO₂ exhibit remarkable electrocatalytic activity toward OER, and the former is more active than the latter. For example, at 1.525 V, the electrocatalytic activity of 6H-SrIrO₃ (measured by the current densities per the geometric area) is almost four times higher than that of IrO₂. Additionally, 6H-SrIrO₃ produces a current density of 10 mA cm⁻²_geo at an overpotential (η) of 248 mV, whereas IrO₂ gives the same current density at a higher overpotential (300 mV). The results also reveal that 6H-SrIrO₃ is among

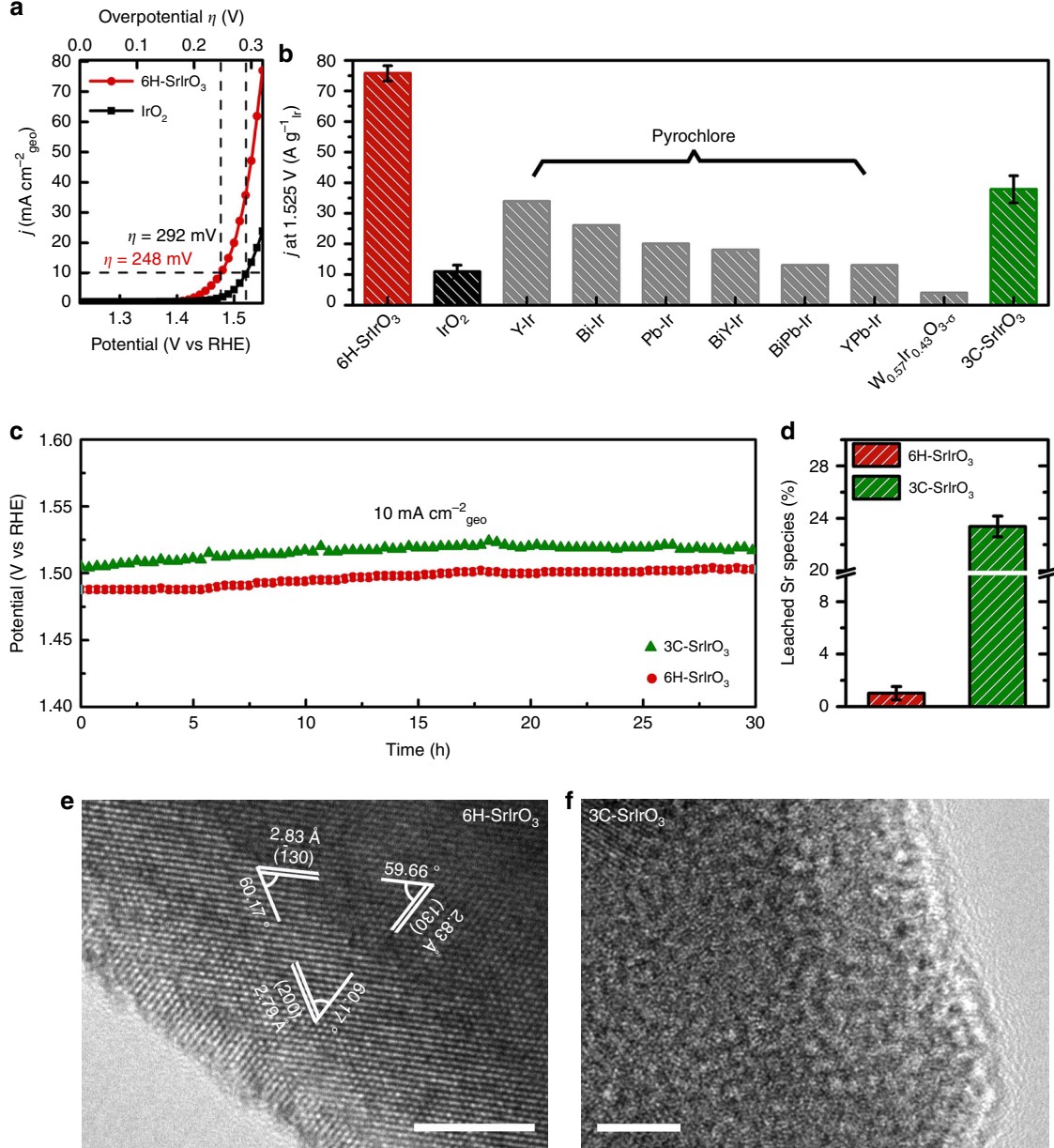

**Fig. 3** Electrocatalytic properties for OER of 6H-SrIrO$_3$. **a** Polarization curves of 6H-SrIrO$_3$ and IrO$_2$ in 0.5 M H$_2$SO$_4$ solution with 85% iR-compensations. The current densities are normalized by the geometric area. **b** Comparison of mass activity, normalized by the mass of iridium at 1.525 V vs. RHE, of 6H-SrIrO$_3$, IrO$_2$, 3C-SrIrO$_3$, and some Ir-containing catalysts reported recently. The error bar represents standard deviation based on five measurements. **c** Chronopotentiometric curves for OER in the presence of 6H-SrIrO$_3$ and 3C-SrIrO$_3$ in 0.5 M H$_2$SO$_4$ solution at 10 mA cm$^{-2}_{geo}$ (without iR compensations). **d** Percentage of total Sr content leached (or deviated from the stoichiometry) in the solution after 30 h-long catalytic stability test in the presence of 6H-SrIrO$_3$ and 3C-SrIrO$_3$. **e, f** HRTEM images for 6H-SrIrO$_3$ and 3C-SrIrO$_3$ after 30 h-long catalytic stability test. Scale bars, 5 nm

the most active Ir-based oxide electrocatalysts for OER in acid (see Supplementary Table 2), although there might be the differences in protocols, such as catalyst loading, use of binder and electrolyte, employed in their electrocatalytic tests. For example, some Pb-, Bi-, or Y-containing, Ir-based pyrolore compounds usually give a current density of 10 mA cm$^{-2}_{geo}$ at overpotantials higher than 340 mV[19–22], and Ir–W bimetallic oxide catalysts with an optimal composition require an over-potential of 370 mV for the acidic OER[22]. Moreover, a perovskite-type La$_2$LiIrO$_6$ catalyst can produce a current density of 10 mA cm$^{-2}_{geo}$ at an overpotential of ca. 300 mV[17], but this material suffers from the serious surface reconstruction, resulting in the generation of IrO$_2$ nanoparticles, during electrocatalysis.

Besides exhibiting good catalytic activity per geometric area, 6H-SrIrO$_3$ also shows good mass activity (normalized by the mass of iridium) for the acidic OER. As shown in Fig. 3b, 6H-SrIrO$_3$ gives the highest mass activity for OER in acid among the iridium-based oxide electrocatalysts reported recently[15,16,18–22]. The mass activity of 6H-SrIrO$_3$ is about seven times as high as that of IrO$_2$, and is 2.2–5.8 times higher than those of Ir-based pyrolore compounds. The mass activity of 6H-SrIrO$_3$ is even around twenty times higher than the Ir–W bimetallic oxide catalyst. Moreover, it is worth noting that the superior catalytic activity of 6H-SrIrO$_3$ does not correlate with its surface area. The surface area (obtained with N$_2$ adsorption method) of 6H-SrIrO$_3$ is typically lower by more than an order of magnitude in

comparison with the Ir-based catalysts mentioned above (see Supplementary Table 2). For example, the $IrO_2$ sample has a larger BET surface area ($19.8 \, m^2 \, g^{-1}$) than $6H\text{-}SrIrO_3$, due to the former's smaller particle size (see Supplementary Figures 4 and 9). This is in agreement with the effective electrochemical active surface area of $IrO_2$, which is measured to be about three times larger than that of $6H\text{-}SrIrO_3$ (Supplementary Figure 10).

**Comparison of electrocatalytic properties for OER between $6H\text{-}SrIrO_3$ and $3C\text{-}SrIrO_3$.** In order to further assess the catalytic properties of $6H\text{-}SrIrO_3$, we synthesized 3C-phase $SrIrO_3$ perovskite ($3C\text{-}SrIrO_3$) as a reference material for comparative studies. $3C\text{-}SrIrO_3$ adopts a well-known pseudo-cubic structure, in which all the $IrO_6$ octahedra are corner-shared (see its crystal structure in Supplementary Figure 11)[29]. Additionally, the thinfilms of $3C\text{-}SrIrO_3$ were recently demonstrated to give the best intrinsic catalytic activity for OER in acid[16], due to the in situ generation of amorphous $IrO_x$ on the surface by serious strontium leaching during electrocatalysis.

The experimental procedures are almost the same for the synthesis of $6H\text{-}SrIrO_3$ and $3C\text{-}SrIrO_3$ (see experimental details in Methods). In order to explore the reasons why the addition of more critic acid in the synthesis can favor the formation of metastable $3C\text{-}SrIrO_3$ over $6H\text{-}SrIrO_3$, we first compared the Ir4$f$ XPS spectra of $IrO_2$, $6H\text{-}SrIrO_3$ and $3C\text{-}SrIrO_3$ (Supplementary Figure 12a). The results reveal that the Ir $4f_{5/2}$ and Ir $4f_{7/2}$ XPS peaks for $3C\text{-}SrIrO_3$ appear at the higher binding energies compared with $IrO_2$ and $6H\text{-}SrIrO_3$. This demonstrates that the average oxidation state of iridium for $3C\text{-}SrIrO_3$ is higher than 4+[30]. The higher average oxidation state of iridium are believed to be favorable for the formation of $3C\text{-}SrIrO_3$. This argument is mainly based on recent studies, in which low-valent heteroatom doping (e.g., Co, Mg, Zn, etc.) can lead to the increase in the oxidation state of iridium and the stabilization of $3C\text{-}SrIrO_3$[28,31,32]. In view of the difference in synthetic systems between our work and the previous studies[28,31,32], we attempted to synthesize 3C-phase $SrIrO_3$ by low-valent heteroatom doping (cobalt as dopants used in this work) with the experimental procedures that were used for the synthesis of $6H\text{-}SrIrO_3$ (see experimental details in Methods). The results show that in this case, the formation of the 3C-phase is indeed preferable, and the resulting material also has a higher oxidation state of iridium (Supplementary Figure 12b). Next, we compared the thermogravimetric (TG) and differential thermal analysis (DTA) in air of the precursors of $6H\text{-}SrIrO_3$ and $3C\text{-}SrIrO_3$. As shown in Supplementary Figure 13a, the organic components in the precursors are oxidatively decomposed in air at elevated temperature (ca. 500 °C), and this process is highly exothermic. The presence of more critic acid (i.e., the precursor for the synthesis of $3C\text{-}SrIrO_3$) is also found to result in more heat released at elevated temperature. The more released heat in the synthesis should favor the generation of high-valent iridium species. This is confirmed by the XPS results (Supplementary Figure 13b), which show that the sample obtained by calcining the precursor of $6H\text{-}SrIrO_3$ at 500 °C contains zero-valent and four-valent Ir species, whereas the sample obtained by calcining the precursor of $3C\text{-}SrIrO_3$ at 500 °C contains four-valent and five-valent Ir species. Overall, the presence of more critic acid favors the formation of $3C\text{-}SrIrO_3$ by facilitating the generation of the more oxidized iridium species.

The $3C\text{-}SrIrO_3$ material we synthesize contains of micron-sized particles with a small surface area comparable with that of $6H\text{-}SrIrO_3$ (Supplementary Figures 14, 15 and Table 2). The $3C\text{-}SrIrO_3$ material has a Sr:Ir atomic ratio of 0.82:1 and a certain amount of $IrO_x$ on the surface. The Sr deficiency for $3C\text{-}SrIrO_3$ is because the acid treatment is necessary for the removal of the $SrCO_3$ impurities during the material synthesis, and simultaneously, the acid treatment also results in some strontium leached from $3C\text{-}SrIrO_3$. This result is also in agreement with the previous study showing the easy strontium leaching of $3C\text{-}SrIrO_3$ in acid [16].

The electrocatalytic result (Supplementary Figure 16) reveals that $3C\text{-}SrIrO_3$ affords a current density of $10 \, mA \, cm^{-2}_{geo}$ at an overpotential ($\eta$) of ~270 mV, which is similar to that (270–290 mV) required by the thin films of $3C\text{-}SrIrO_3$ reported previousdly[16]. Obviously, $6H\text{-}SrIrO_3$ has a higher catalytic activity than $3C\text{-}SrIrO_3$ because the former needs a smaller overpotential of ~248 mV to deliver a current density of $10 \, mA \, cm^{-2}_{geo}$. The superior catalytic activity of $6H\text{-}SrIrO_3$ is further supported by the result that the former exhibits two times mass activity as high as the later (Fig. 3b).

Moreover, both $6H\text{-}SrIrO_3$ and $3C\text{-}SrIrO_3$ exhibit good catalytic stability for OER in acid, as shown in Fig. 3c. However, after 30 h-long electrocatalysis test, only ~1% of total Sr content (0.9–1.5 Sr layers of $6H\text{-}SrIrO_3$) is detected in the acidic solution in the presence of $6H\text{-}SrIrO_3$, whereas ~24% of total Sr content is deviated from the stoichiometry of $3C\text{-}SrIrO_3$ due to the Sr leaching during electrocatalysis (Fig. 3d). Considering that $3C\text{-}SrIrO_3$ has been a Sr-deficient material with an 18% deviation before the electrocatalysis, the $3C\text{-}SrIrO_3$ material after electrocatalysis has a total strontium deficiency up to 42%. The result is also consistent with the 30–50% of total Sr content leaching for the thin films of $3C\text{-}SrIrO_3$ as oxygen evolution catalysts in acid[16]. Additionally, HRTEM and X-ray photoelectron spectroscopy (XPS) results show that $6H\text{-}SrIrO_3$ has good structural stability and keeps high crystallinity (Fig. 3e and Supplementary Figure 17), but $3C\text{-}SrIrO_3$ takes place obvious surface amorphourization during electrocatalysis (Fig. 3f). This is further supported by the high-angle annular dark field (HAADF) STEM image (Supplementary Figure 20), which shows that there is not secondary, amorphous $IrO_x$ phase generated on the $6H\text{-}SrIrO_3$ surface after OER. The $N_2$ isotherm data of $6H\text{-}SrIrO_3$ (Supplementary Figure 18) reveal that there are not micropores formed after OER. Moreover, the comparison of the $6H\text{-}SrIrO_3$'s CV curves before and after OER (Supplementary Figure 19) shows that the CV shape of $6H\text{-}SrIrO_3$ after OER is not exactly same as that before OER, but does not change a lot. This suggests that $6H\text{-}SrIrO_3$ might undergo a weak surface reconstruction during OER probably thanks to the slight Sr leaching during OER. It should be pointed out that while the better structural stability of $6H\text{-}SrIrO_3$ than $3C\text{-}SrIrO_3$ has been unambiguously confirmed, the possibility of a very thin amorphous $IrO_x$ layer (or a tiny amount of $IrO_x$ clusters) on the $6H\text{-}SrIrO_3$ surface cannot be completely ruled out at current stage.

The better structural stability of $6H\text{-}SrIrO_3$ in comparison with $3C\text{-}SrIrO_3$ can be explained by the comparison of their O K-edge X-ray absorption spectra (Supplementary Figure 21). The comparison reveals that there is stronger Ir $5d$-O $2p$ hybridization or stronger Ir–O covalence for $3C\text{-}SrIrO_3$. The stronger covalence results in the easier structural loss of the perovskite phase or surface amorphourization during electrocatalysis, as suggested by the recent studies on perovskite electrocatalysts [33].

In order to further assess the stability of $6H\text{-}SrIrO_3$, we compared the Ir leaching during OER in the presence of $6H\text{-}SrIrO_3$ and $IrO_2$. We used inductively coupled plasma atomic emission spectroscopy (ICP-OES) to detect the Ir species in the electrolyte at a current density of $10 \, mA \, cm^{-2}_{geo}$ with $6H\text{-}SrIrO_3$ or $IrO_2$ as the electrocatalyst. The result (Supplementary Figure 22) reveals that a tiny amount of leached

Ir species are detected in the electrolytes for both 6H-SrIrO₃ and IrO₂ as the electrocatalysts, but the amount of leached Ir species in the presence of 6H-SrIrO₃ as the electrocatalyst is obviously lower than that in the presence of IrO₂ as the electrocatalyst. This result further demonstrates that 6H-SrIrO₃ is a stable electrocatalyst for OER in acid.

In order to determine the Faradaic efficiency during the OER, we compared the amount of $O_2$ produced from the OER in the presence of 6H-SrIrO₃ with the theoretically expected $O_2$ amount that can be generated from the OER. The result (Supplementary Figure 23) reveals that the detected $O_2$ amount is very close to the expected value during the OER, further demonstrating the about 100% Faradaic efficiency given by 6H-SrIrO₃.

## Discussion

We performed density functional theory (DFT) computations in order to get a better understanding of the electrocatalytic activity of 6H-SrIrO₃ for OER. We constructed the correlative theoretical models in view of the above experimental results, including the {001} facet of 6H-SrIrO₃ as the main exposed one and a small amount of surface strontium leached during electrocatalysis. We cleaved the optimized 6H-SrIrO₃ bulk structure through its (001) plane to obtain two representative surfaces (Fig. 4a–d). The surface-I exposes the Ir atoms in the face-sharing octahedral dimers (Fig. 4a, b), while the surface-II exposes the Ir atoms in the isolated, corner-sharing $IrO_6$ octahedra (Fig. 4c, d). On the basis of these surface models, we investigated the electrocatalytic activity for OER of 6H-SrIrO₃ according to the approach proposed by Rossmeisl et al.[34,35] In this approach, the OER is suggested to include the four elementary reaction steps and each step involves one proton/one electron-coupled transfer process, as

shown below:

$$H_2O + * \rightarrow HO^* + H^+ + e^- \qquad (1)$$

$$HO^* \rightarrow O^* + H^+ + e^- \qquad (2)$$

$$O^* + H_2O \rightarrow HOO^* + H^+ + e^- \qquad (3)$$

$$HOO^* \rightarrow * + O_2 + H^+ + e^- \qquad (4)$$

where * and X* (X = HO, O, or HOO) represent a surface active site and an adsorbed X intermediate on the surface, respectively.

As shown in Fig. 4e and f, all of the four elementary reaction steps involved in the OER on both surface-I and surface-II are found to move uphill in free energy when no potential is applied ($U = 0$ V). An additional potential has to be applied to make every step go downhill in free energy, and the minimum additional potentials are determined to be 1.69 and 1.80 V for the surface-I and surface-II, respectively. Considering the equilibrium potential of the OER ($U = 1.23$ V), the minimum theoretical overpotentials ($\eta$) are calculated to be 0.46 and 0.57 V for the surface-I and surface-II, respectively. These results further demonstrate that the surface-I, comprising the Ir atoms in the face-sharing octahedral dimers, has better catalytic activity for OER than the surface-II with the Ir atoms in the isolated, corner-sharing $IrO_6$ octahedra. For comparison, we also explore the OER activities on the (110) surface of $IrO_2$ and (010) surface of 3C-SrIrO₃, and both the sampled surfaces have the similar structural feature to the surface-II, as illustrated in Supplementary Figures 24 and 25. Our computed overpotentials are 0.59 and 0.56 V for $IrO_2$ and 3C-SrIrO₃, respectively, both of which are

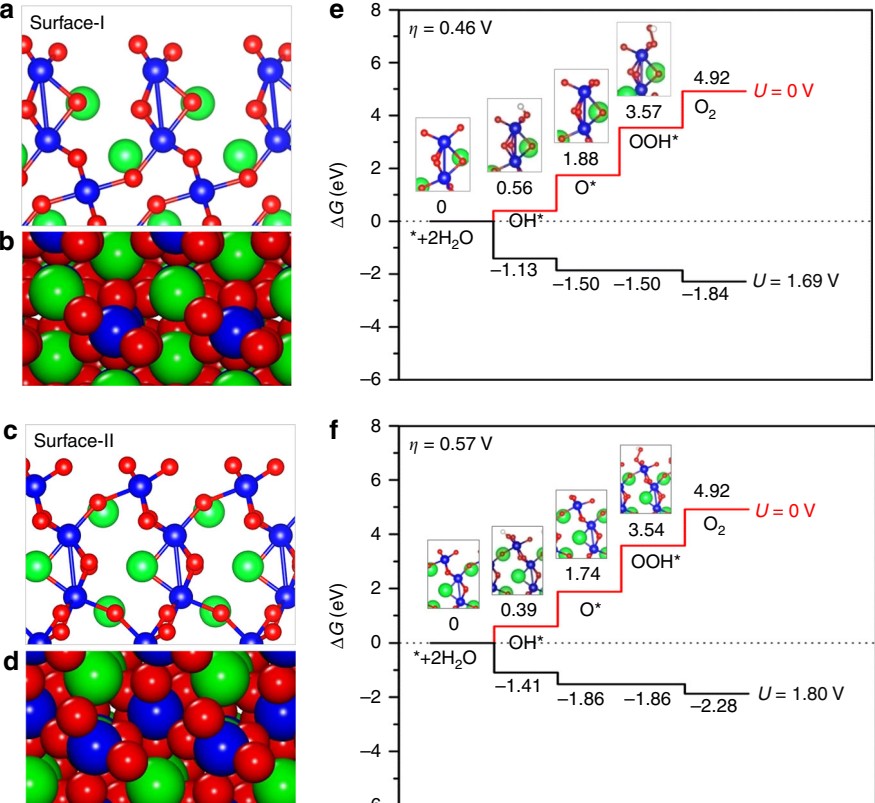

**Fig. 4** Theoretical understanding of electrocatalytic activity for OER of 6H-SrIrO₃. **a** Side and **b** top views of the surface-I of 6H-SrIrO₃; **c** Side and **d** top views of the surface-II of 6H-SrIrO₃; free-energy diagrams of four elementary reaction steps for the OER on **e** the surface-I and **f** surface-II of 6H-SrIrO₃ at the different applied potentials. The optimized structures of HO, O, and HOO adsorptions on the surface-I and surface-II are also shown in **e** and **f**

comparable to that of the surface-II (0.57 V) and larger than that of surface-I (0.46 V). It is worth mentioning that our computed overpotential on the (110) surface of $IrO_2$ is also close to the previously reported one[34]. The above calculations imply that the existence of face-sharing octahedral dimers (the main structural feature of surface-I) is mainly responsible for the superior OER activity of 6H-$SrIrO_3$, as observed in the above experimental results.

From the Fig. 4e and f, we can also find that the potential-determining step for both surface-I and surface-II is the third elementary reaction step involved in the OER (i.e., $O^\star + H_2O \rightarrow HOO^\star + H^+ + e^-$) because the maximum Gibbs free-energy difference is between $\Delta G_{HOO^\star}$ and $\Delta G_{O^\star}$. The facilitated potential-determining step involved in the OER for the surface-I, with respect to the surface-II, can be attributed to the relatively weak O binding (or the relatively low ability of Ir atoms to bind oxygen) on the surface-I, because the surface-I binds oxygen with an $O^\star$ binding energy of 1.848 eV, while the surface-II binds $O^\star$ more strongly ($\Delta E_{O^\star} = 1.711$ eV).

When DFT calculations have provided some useful insights into the catalytic active sites and the activity trend for the Ir-based materials from the theoretical computations is consistent with that from our experiments, there is difference between the thermodynamic overpotentials obtained by DFT and the experimental ones. This difference might be caused by multiple factors. First, all the computations are based on thermodynamic considerations. Other important electrocatalytic processes, such as mass and charge transport, cannot be involved in the DFT calculations. Second, the distribution and concentration of active sites on the real material surface are generally very complex, but this complexity cannot be adequately reflected by DFT calculations. Third, at current stage, DFT calculations are not capable of providing atomic information on the more complex catalyst-electrolyte interface, which is crucially important for the electrocatalysis process.

In summary, 6H-$SrIrO_3$ has been identified as a highly efficient, low-iridium oxygen evolution electrocatalyst that can work well in acidic media. 6H-$SrIrO_3$ is an unusual perovskite oxide electrocatalyst with face-sharing $IrO_6$ octahedral dimers. The unique face-sharing $IrO_6$ octahedral subunits of 6H-$SrIrO_3$ are proven to be important for its remarkable catalytic activity and structural stability. These good properties also make 6H-$SrIrO_3$ a promising electrocatalyst for proton exchange membrane (PEM) water electrolysis. The findings in this work might help spark a kind of creativity in the rational design/synthesis of high-performance electrocatalysts for the acidic OER.

## Methods

**Chemicals and reagents**. Potassium hexachloroiridate(IV) ($K_2IrCl_6$, 99.99%) was purchased from Aladdin. Strontium nitrate ($Sr(NO_3)_2$), citric acid monohydrate ($C_6H_8O_7 \cdot H_2O$), ethylene glycol, hydrochloric acid (HCl) and sulfuric acid ($H_2SO_4$) were purchased from Beijing Chemical Factory. Cobalt nitrate hexahydrate (Co $(NO_3)_2 \cdot 6H_2O$) was purchased from Shantou Xilong Chemical Factory. Nafion® perfluorinated resin solution (5 wt.% in mixture of lower aliphatic alcohols and water containing 45% water) was purchased from Sigma-Aldrich. All the chemicals and reagents were used without further purification. Highly purified water (>18 MΩ cm resistivity) was provided by a PALL PURELAB Plus system.

**Materials synthesis**. For the synthesis of 6H-$SrIrO_3$, $Sr(NO_3)_2$ (280 mg) and citric acid (280 mg) were mixed with 5 mL deionized $H_2O$ to form the solution 1, and $K_2IrCl_6$ (80 mg) was mixed with 4 mL ethylene glycol to form the solution 2. The solution 1 was then stirred into the solution 2 dropwise. The resulting mixture was dried at 150 °C for 12 h to obtain a brown solid product as the precursor, followed by calcination in air at 200 °C (6 h), 300 °C (6 h), 500 °C (3 h), and 700 °C (6 h) in succession. The heating rate was 1.7 °C $min^{-1}$. The obtained solid product containing $SrIrO_3$ and $SrCO_3$ was treated with 1 M HCl for 6 h to remove the $SrCO_3$ impurities, giving the 6H-$SrIrO_3$ material. The sequence of synthesis steps was refined through trials and errors.

In order to further assess the stability of 6H-$SrIrO_3$ in 1 M HCl solution, we dispersed the as-obtained 6H-$SrIrO_3$ powders in 1 M HCl solution for 2 days, and then used ICP-OES to detect the Sr and Ir ions in the solution. The result shows that there is no detectable leached Sr and Ir species in the solution, demonstrating the good stability of 6H-$SrIrO_3$ in 1 M HCl solution.

For the synthesis of 3C-$SrIrO_3$, the experimental procedures are almost the same as these for the synthesis of 6H-$SrIrO_3$, except the amount of citric acid (840 mg) that was used to form the solution 1.

For the synthesis of Co-doped 3C-$SrIrO_3$ as a control sample, the experimental procedures are almost the same as these for the synthesis of 6H-$SrIrO_3$. The only difference was that a certain amount of cobalt nitrate hexahydrate (14.5 mg) was added to form the solution 2. In the resulting sample, labeled as Co-doped 3C-$SrIrO_3$, the Co:Ir atomic ratio is about 1:4.

In order to synthesize the $SrIrO_3$ phases (6H-$SrIrO_3$ and 3C-$SrIrO_3$) without $IrO_2$ as the secondary phase, an excessive amount of strontium (Sr:Ir atomic ratio = 8:1) was used during the synthesis in this work. The use of the excessive amount of strontium resulted in the formation of strontium carbonates ($SrCO_3$) as the impurities (please see Supplementary Figure 26). The acid treatment with 1 M HCl was employed to remove the $SrCO_3$ impurities.

For the synthesis of $IrO_2$, the experimental procedures are almost the same as these for the synthesis of 6H-$SrIrO_3$. The only difference was that $Sr(NO_3)_2$ was not involved in the synthesis. Supplementary Figure 27 presents XRD patterns of the $IrO_2$ samples obtained by calcining the precursor at 300, 500, and 700 °C.

**Characterizations**. Powder X-ray diffraction patterns (PXRD) of the samples were collected with an X-ray diffractometer (RIGAKU, Japan, model D/MAX2550 V/ PC) with a scan speed of 1° $min^{-1}$ within the diffraction angle range from 10 to 140°. Profile fitting for the PXRD pattern was performed with general structure analysis system (GSAS) program. The scanning electron microscope (SEM) images were obtained with a JEOL JSM 6700F electron microscope. The transmission electron microscope (TEM) images were obtained with a Philips-FEI Tecnai G2S-Twin microscope equipped with a field emission gun operating at 200 kV. Inductively coupled plasma atomic emission spectroscopy (ICP-OES) was performed on a Perkin-Elmer Optima 3300DV ICP spectrometer. The X-ray photo-electron spectroscopy (XPS) was performed on an ESCALAB 250X-ray photoelectron spectrometer with a monochromatic X-ray source (Al Kα $h\nu$ = 1486.6 eV). The O K-edge X-ray absorption near-edge (XANES) spectra were measured in the total electron yield mode in a vacuum chamber ($<5 \times 10^{-8}$ Pa). This measurement was performed at the BL12B-a beamline of National Synchrotron Radiation Laboratory (NSRL) in China. The electrical resistivity was investigated with Van Der Pauw method, and the measurement was conducted in a Physical Property Measurement System (PPMS, Quantum Design). High-angle annular dark field scanning transmission electron microscopy (STEM) images were acquired using an aberration-corrected STEM with the model of FEI Titan Cubed Themis G2 300, whose accelerating voltage and electron current were set at 200 kV and around 40 pA, respectively. (Note that because 6H-$SrIrO_3$ is very sensitive to the electron beams, high-angle annular dark field STEM images must be obtained at the lowest electron current of 40 pA).

**Electrochemical measurements**. All the electrochemical measurements were performed with a three-electrode system by using a CHI instrument (Model 650E). The electrolyte was 0.5 M $H_2SO_4$, and was bubbled with $O_2$ gas during the electrochemical measurements. A carbon rod was used as the counter electrode, and a saturated calomel electrode (SCE) was used as the reference electrode. The saturated calomel electrode was calibrated by the reversible hydrogen electrode (RHE), giving their conversion equation: $E_{vs.RHE} = E_{vs.SCE} + 0.241$ V. The scan rate was 0.5 mV $s^{-1}$ when linear sweep voltammetry (LSV) was used. The data in this work were compensated by 85% iR-drop. The geometric surface area of the electrode was used to normalize the current density.

For the preparation of working electrodes, the powdered catalyst (7 mg), including 6H-$SrIrO_3$, 3C-$SrIrO_3$, and $IrO_2$, was dispersed in a mixture comprising isopropanol (100 μL) and Nafion solution (10 μL, 5 wt.% in mixture of lower aliphatic alcohols and water). 1 μL of the resulting mixture was then drop-casted onto a glassy carbon electrode with a diameter of 3 mm for drying. The working electrode had a catalyst loading of ca. 0.90 mg $cm^{-2}$.

The electrochemically active surface area (ECSA) of each sample was obtained by determining the double-layer capacitance at non-Faradaic potential range, according to the method reported by Jaramillo et al.[36] A series of cyclic voltammetry (CV) measurements were performed first at various scan rates (10, 25, 50, 75 and 100 mV $s^{-1}$) in the potential window between 0.83 and 0.93 V vs. RHE. Then, a linear plot was obtained by establishing the relationship between the difference of the anodic and cathodic currents ($i_a - i_c$) at 0.88 V vs. RHE and the scan rate. The double-layer capacitance ($C_{dl}$) is one half of the slope value of the fitting line. The ECAS can be obtained by dividing $C_{dl}$ by a specific capacitance ($C_s$). A $C_s$ value of 0.035 mF $cm^{-2}$ was suggested in the previously reported work [36].

The reference electrode (i.e., saturated calomel electrode, SCE) was calibrated by the reversible hydrogen electrode (RHE) in 0.5 M $H_2SO_4$. Two Pt electrodes were first cleaned by cycled in 0.5 M $H_2SO_4$ between −2 and 2 V for 2 h, and then used as the working electrode and the counter electrode, respectively. During the calibration, the electrolyte was saturated by $H_2$. A series of controlled-potential chronoamperometric curves were carried out around the possible zero current potential (the interconversion between the hydrogen oxidation and hydrogen

evolution reaction) with a dwell time of 5 min. The result showed that the potential of zero net current was found at $-0.241$ V vs. the SCE electrode. And thus, the potentials, measured against SCE, were converted into the potentials vs. RHE by using the equation: $E_{vs.RHE} = E_{vs.SCE} + 0.241$ V.

**Computation details.** The generalized gradient approximation (GGA) with the Perdew–Burke–Ernzerhof exchange-correlation functional[37] (including a semi-empirical van der Waals (vdW) correction to account for the dispersion interactions)[38,39] and a 400-eV cutoff for the plane-wave basis set are employed to perform all the density functional theory (DFT) computations within the frame of Vienna ab initio simulation package (VASP)[40,41]. The projector-augmented plane wave (PAW) is used to describe the electron-ion interactions[42,43]. $5 \times 3 \times 2$ and $5 \times 5 \times 1$ Monkhorst-Pack grid $k$-points are employed for geometric optimization of 6H-SrIrO$_3$ and the corresponding slab surfaces, respectively. The density of state (DOS) of the 6H-SrIrO$_3$ is computed by using 77 $k$-points. The convergence threshold is set as $10^{-4}$ eV in energy and 0.05 eV/Å in force. For all the calculations of slab models, the symmetrization is switched off and the dipolar correction is included. Our computed lattice parameters for 6H-SrIrO$_3$ are about $a = 5.554$, $b = 9.584$, and $c = 14.024$ Å, all of which are in good agreement with their corresponding experimental values [28].

The free energy of H$^+$ + e- can be half of formation energy of H$_2$ at 298 K and 1 atm. The free energy of the OER is computed by the equation $\Delta G = \Delta E + \Delta ZPE - T\Delta S$. The value of $\Delta E$ is obtained by the computation of geometrical structures. The values of $\Delta ZPE$ and $\Delta S$ are determined by employing the computed vibrational frequencies and standard tables for the reactants and products in the gas phase[44]. The entropy for the adsorbed atoms/molecules at the surface active site are assumed to be zero. The temperature dependence of the enthalpy is neglected in the calculations. Moreover, an external bias $U$ is imposed on each step by including a $-eU$ term in the computation of reaction free energy. Consequently, the reaction free energy of each step can be expressed as follows:

$$\Delta G_1 = E(\text{HO}^*) - E(^*) - E_{\text{H}_2\text{O}} + 1/2E_{\text{H}_2} + (\Delta ZPE - T\Delta S)_A - eU \quad (5)$$

$$\Delta G_2 = E(\text{O}^*) - E(\text{HO}^*) + 1/2E_{\text{H}_2} + (\Delta ZPE - T\Delta S)_B - eU \quad (6)$$

$$\Delta G_3 = E(\text{HOO}^*) - E(\text{O}^*) - E_{\text{H}_2\text{O}} + 1/2E_{\text{H}_2} + (\Delta ZPE - T\Delta S)_C - eU \quad (7)$$

$$\Delta G_4 = E(^*) - E(\text{HOO}^*) + E_{\text{O}_2} + 1/2E_{\text{H}_2} + (\Delta ZPE - T\Delta S)_D - eU \quad (8)$$

where $E(^*)$, $E(\text{HO}^*)$, $E(\text{O}^*)$, and $E(\text{HOO}^*)$ are the computed DFT energies of the pure surface and the adsorbed surfaces with HO\*, O\*,and HOO\*, respectively. $E_{\text{H}_2\text{O}}$, $E_{\text{H}_2}$, and $E_{\text{O}_2}$ are the computed energies for the sole H$_2$O, H$_2$ and O$_2$ molecules, respectively. For the total reaction H$_2$O → 1/2O$_2$ + H$_2$, the free-energy change is fixed at the experimental value of 2.46 eV per water molecule. When forming one molecule of O$_2$ in the reaction step, the reaction free energy can be expressed as $\Delta G_{(2\text{H}_2\text{O}\rightarrow\text{O}_2+2\text{H}_2)} = 4.92$ eV $= E_{\text{O}_2} + 2E_{\text{H}_2} - 2E_{\text{H}_2\text{O}} + (\Delta ZPE - T\Delta S)_{(2\text{H}_2\text{O}\rightarrow\text{O}_2+2\text{H}_2)}$. The reaction overpotential can be obtained by evaluating the difference between the minimum voltage needed for the OER and the corresponding voltage needed for changing all the free-energy steps into downhill. The correlative theoretical models are constructed based on the experimental results, including the {001} facet of 6H-SrIrO$_3$ as the main exposed one and a small amount of surface strontium leached during electrocatalysis. The dissolution of SrIrO$_3$ to form Sr$^{2+}$ can be written as SrIrO$_3$ + 2 H$^+$ → IrO$_2$ + Sr$^{2+}$ + H$_2$O, and in this situation, Sr atoms and the related O atoms in the uppest layer for these structural models are discarded from a stoichiometric SrIrO$_3$ (001) surface. All the theoretical models have the thickness of six layers. During the computational process, the upper two layers in the model are fully relaxed without any symmetry or directional restrictions, while the remaining four layers are kept frozen.

The ELF was extensively employed to understand the localized bonding character in the crystal structures[26,27]. The ELF can be described in the form of a contour plot in real space with values ranging from 0 to 1. The region close to 1 indicates the presence of strong covalent electrons or lone-pair electrons. While the region near to 0.5 represents homogenous electron gas or the presence of metallic bonds, the region close to 0 means a low electron density area. The ELF is different from the molecular orbital theory that can display the orbital orientation. In our work, we employed the ELF method to confirm the existence of Ir–Ir metallic bond in the face-shared IrO$_6$ dimers. As presented in Fig. 1d, the correlative ELF values with a range of 0.4 to 0.5 distribute between the two neighboring Ir atoms in face-sharing IrO$_6$ dimers, indicating that Ir–Ir bond is metallic in character.

## Data availability
The authors declare that the data supporting the findings of this study are available within the paper and its supplementary information files.

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

## Acknowledgements

This work is financially supported by the National Natural Science Foundation of China (NSFC, Grant Nos. 21771079, 21673094, 21673093, and 21573090), National Key R&D Program of China, Grant No. 2017YFA0207800, Jilin Province Science and Technology Development Plan 20170101141JC and 20170101175JC, Program for JLU Science and Technology Innovative Research Team (JLUSTIRT), Fok Ying Tung Education Foundation, Grant No. 161011 and Science and Technology Research Program of Education Department of Jilin Province (JJKH20170780KJ). We thank the NSRL beamline BL12B-a of National Synchrotron Radiation Laboratory for the XAS measurement, Center for Electron Microscopy in Tianjin University of Technology, and the Computing Center of Jilin Province for the supercomputer time. We thank the National Natural Science Foundation of China (Grant No. 21621001) and the 111 Project (No.B17020) for financial support.

## Author contributions

X.Z. conceived the idea, organized the data, and wrote the manuscript. G.Y. and W.C. performed the theoretical computations and analyzed the theoretical results. L.Y. synthesized the materials and studied their electrochemical properties. X.A., X.L., and J.-S.C. assisted L.Y. with the materials synthesis and the analysis of crystal structure. W.Y. and H.D. performed the XAS measurement. T.W., C.Z., and X.H. assisted W.C. with DFT calculations. All of the authors have read the manuscript and agree with its content.

## Additional information

**Competing interests:** The authors declare no competing interests.

