## [Peer Review File · Nature Communications]

Reviewers' comments:

Reviewer #1 (Remarks to the Author):

In this work, Yang et al studied the OER activity and stability of the hexagonal 6H SrIrO₃ phase which they find are better than for cubic SrIrO₃ or state-of-the-art IrO₂. The results are of great interests and could be potentially published in a journal such as Nature Communications. Nevertheless, several details remain to be clarified in order to grant publication for this work, especially concerning the physical characterizations of the phase and the type of bonding as well as clear comparison in terms of stability and Ir leaching when compared to IrO₂.

The discussion about the type of bonding found for the 6H SrIrO₃ appears as insufficient and the measurements presented in the manuscript don't support the claims. Indeed, it is not clear how the ELF function presented Figure 1 help at visualizing the Ir-Ir metallic bonding since it appears that no Ir orbitals are pointing towards each other (only orbitals participating to Ir-O bonds appear on the ELF function Figure 1). Moreover, while I generally agree that Ir-O bond are often weaker for face-sharing octahedral, the claim made by the authors in the text is not supported by any measurements. Therefore, additional measurements such as XAS at the O K-edge or measurements at the Ru edge would be necessary to demonstrate greater covalency of the Ir-O bond in the hexagonal form of SrIrO₃ when compared to its cubic form. Hence, the discussion page 12 about the different bonding for the 6H SrIrO₃ being responsible for its better stability is not supported.

I am surprised to see atomic distances in Figure 1 while in Figure 2 no Rietveld analysis was performed on the XRD obtained after synthesis. The authors must perform such analysis in order to discuss the atomic distances and make sure that the material they prepare is indeed 6H SrIrO₃ with similar crystallographic properties than the ones reported in the literature. Same applies to the electrical conductivity that the authors did not measure in this work but rather infer from previous works. Caution must be exercised since slight differences in oxygen content or atomic arrangements due to the synthesis protocol can drastically modify these properties.

Concerning the chemical stability of the phase in 0.5M H₂SO₄, when the authors claim that "only" 1.3% of the Ir is leached out under these conditions, this value is not negligible. In fact, it would be interesting to compare this value with the one obtained for IrO₂. Indeed, it seems very high especially when compared to value often measured for IrO₂ (see work from Mayrhofer and others for comparison). Same applies for Ir leaching during OER, it should be compared with IrO₂ or the value obtained should be discussed with what was previously reported for IrO₂. Only providing this comparison will allow for stating that 6H SrIrO₃ is indeed more stable than other catalysts.

Concerning the OER activity, the authors should avoid statement such as "the most active Ir-based oxide electrocatalysts for OER in acid" page 9. Indeed, protocols for measuring the different phases might be different (such as loading, use of binder or others). Same applies for the statement concerning the mass activity.

Page 10, when describing the "as-synthesized" cubic SrIrO₃ that has a deficiency of Sr due to acid leaching what the authors mean by as-synthesized? The EDX has been taken after synthesis, or after acid treatment mimicking the one performed by Seitz et al? That should be clarified.

When discussing the thermodynamical potential obtained by DFT, the authors do not discuss why the overpotential computed by DFT is by far greater than the one experimentally obtained (almost 200 mV greater). Where this discrepancy comes from? This should be discussed in greater details. Moreover, in addition to compare the computed overpotential depending on the surface studied for 6H SrIrO₃,

authors should compare directly with the cubic SrIrO₃. Indeed, even if the oxygen atoms are coordinated in a corner-shared fashion in surface II, their energy will be dependent on the subjacent structure.

Page 15, when the authors explain that their DFT study is consistent with their “analysis” of the Ir-O bond being weaker for the face-shared octahedral, this claim is not supported. Indeed, rather than an “analysis”, the authors only discuss in the first part of their manuscript very general “feelings” that are not supported by any measurements. Moreover, when discussing about the nucleophilic attack of water on the oxygen sites, this mechanism has been discussed by Strasser and Grimaud as resulting from the oxidation of surface oxygen under OER conditions, which eventually can result into degradation and instability of the materials. Why would that be different for 6H SrIrO₃? Rather than this chemical acid-base step, it seems that the 6H perovskite behaves following the classical PCET mechanism.

Finally, the methods section is lacking a lot of details, such as the characterizations techniques or purity of precursors or acids etc. This must be fixed in order to allow for good reproducibility of these results.

Additional comments:

Page 9: La₂LiIrO₆ is a perovskite, not a pyrochlore

Numerous typos such as actahedral throughout the text, « structure » page 5, potanitals page 13 etc.

Microsized should be micronsized page 10

Page 9, when referring to the mass activity, it should not be Figure 2b but rather Fig. 3b.

Reviewer #2 (Remarks to the Author):

Yang and coworkers report the OER on 6H-SrIrO₃ and found the OER to be more active than 3C-SrIrO₃, IrO₂, and Iridium-oxide pyrochlores. The search for high-activity OER catalysts is an important direction in energy science. The high OER performance of 6H-SrIrO₃ is therefore a welcome result. However, there are a few technical concerns, which should be fully and thoroughly addressed.

1. Synthesis. Why do the authors choose to use a series of heat treatment (200C for 6h, 300C for 6h, 500C for 3h, and 700C for 6h) for the 6H-SrIrO₃ synthesis? Was this synthesis randomly discovered or was there a rationale behind it?

2. 6H-SrIrO₃ is a stable phase at RT, whereas 3C-SrIrO₃ is not. Why does adding more citric acid favor the metastable 3C-SrIrO₃ formation over 6H-SrIrO₃?

3. Why are the BET areas of 6H-SrIrO₃ and 3C-SrIrO₃ so different from IrO₂ when they were both synthesized using the same method, simply without adding Sr?

4. What impurities are the authors getting rid of with the 1M HCl treatment?

5. The authors argue that the surface structure of 6H-SrIrO₃ is stable because very little Sr leaches out after the OER testing. There are three questions regarding this claim.

a. Could Sr leach out during the 1M HCl treatment? If so, would not the overall Sr loss be greater than the amount of Sr measured in the electrolyte?

b. ~1% Sr dissolution is significant for micron-sized particles and can correspond to a few monolayer dissolutions. The authors should estimate how many monolayers this is.

c. It is difficult to use TEM to quantify the first few surface monolayers. Supplemental methods should be used to verify that the surface structure of 6H-SrIrO₃ is stable.

6. Page 7, the statement that “the exposed facets of micro-sized 6H-SrIrO₃ plates is {001}” should be supported with orientation mapping data from EBSD.

7. Was the reference calibration done using hydrogen redox tested in the same electrolyte? If not, this should be done since small ionic activities inside the reference can shift its true potential.

8. The authors should compare the DFT theoretical overpotential for IrO₂, 6H-SrIrO₃, and 3C-SrIrO₃ in the same figure (Ref. 30 and 31). The authors should also calculate IrO₂ to ensure that their DFT models are consistent with all three materials in order to ensure fair comparisons.

Reviewer #3 (Remarks to the Author):

Review on “Efficient Oxygen Evolution Electrocatalysis in Acid by an Unusual Perovskite Oxide with Face-Sharing IrO₆ Octahedral Dimers”.

The lack of good alternative to Ir based OER catalysts in acidic environments is a huge challenge for electrocatalysis today. The approach in this study is to reduce the amount and increase the activity of Ir, which is a valid approach. The issue of Ir for OER is similar to Pt for ORR, except that OER probably is more important than ORR and Ir is rarer than Pt. Whereas, several different Pt based catalysts for ORR are presented in top journals every year, Ir based OER catalysis is much less studied. Therefore, I find the subject of this paper highly relevant and interesting for Nature Comm.

The activities compared to IrO₂ is per geometric surface area and per mass. The activity per electrochemical surface area is discussed in text. The conclusion is that on all measures the perovskite outperforms IrO₂. This suggests that at least part of the increased activity is related to more active sites on perovskite surface (which the authors also write). This leads to DFT simulations, where the analysis follows ref. 31. In the overpotential the difference is -0.04 V (measured/geo area).

The DFT simulations of the free energy diagrams support the difference between two surfaces, but IrO₂ e.g. (110) was not included. I think free energy diagram of pure IrO₂ should be included in fig 4 for reference.

The reason surface I is found more active than II is due to the weaker oxygen binding and due to the almost constant OOH binding. The difference between surface I and II, does not exactly follow the scaling between OH and OOH. Therefore, O and OH could be in local minima structures, I would like the authors to test that the O and OH structures of surface I cannot be relaxed to slightly more stable structures. This could be done by starting in the OOH structure and remove OH.

When this is done I recommend that the study is published in Nature Comm.

Reviewer's Comments and Our Responses:

Reviewer: 1

Comment 1: In this work, Yang et al studied the OER activity and stability of the hexagonal 6H SrIrO₃ phase which they find are better than for cubic SrIrO₃ or state-of-the-art IrO₂. The results are of great interests and could be potentially published in a journal such as Nature Communications. Nevertheless, several details remain to be clarified in order to grant publication for this work, especially concerning the physical characterizations of the phase and the type of bonding as well as clear comparison in terms of stability and Ir leaching when compared to IrO₂.

Response 1: We thank this reviewer for his/her comments here! We have addressed the comments point-by-point as follows.

Comment 2: The discussion about the type of bonding found for the 6H SrIrO₃ appears as insufficient and the measurements presented in the manuscript don't support the claims. Indeed, it is not clear how the ELF function presented Figure 1 help at visualizing the Ir-Ir metallic bonding since it appears that no Ir orbitals are pointing towards each other (only orbitals participating to Ir-O bonds appear on the ELF function Figure 1). Moreover, while I generally agree that Ir-O bond are often weaker for face-sharing octahedral, the claim made by the authors in the text is not supported by any measurements. Therefore, additional measurements such as XAS at the O K-edge or measurements at the Ir edge would be necessary to demonstrate greater covalency of the Ir-O bond in the hexagonal form of SrIrO₃ when compared to its cubic form. Hence, the discussion page 12 about the different bonding for the 6H SrIrO₃ being responsible for its better stability is not supported.

Response 2: Thanks for the reviewer's valuable comments.

(1) Please let us say sorry for not giving a clear explanation why the electron location function (ELF) can be used to understand the bonding character of Ir-Ir in the face-shared IrO₆ dimers in the original manuscript. Actually, the ELF was extensively employed to understand the localized bonding character in the crystal structures (*e.g.*, *Angew. Chem. Int. Ed.* 1992, **31**, 187; *Phys. Rev. B* 2014, **90**, 035447; *Nano Lett.* 2015, **15**, 1296; *J. Am. Chem. Soc.* 2008, **130**, 5848; *J. Am. Chem. Soc.* 2017, **139**, 12370). The ELF can be described in the form of a contour plot in real space with values ranging from 0 to 1. The region close to 1 indicates the presence of strong covalent electrons or lone-pair electrons. While the region near to 0.5 represents homogenous electron gas or the presence of metallic bonds, the region close to 0 means a low electron density area. The ELF is different from the molecular orbital theory that can display the orbital orientation. In our work, we employed the ELF method to confirm the existence of Ir-Ir metallic bond in the face-shared IrO₆ dimers. As presented in Figure 1d, the correlative ELF values with a range of 0.4 to 0.5 distribute between the two neighboring Ir atoms in face-sharing IrO₆ dimers,

indicating that Ir-Ir bond is metallic in character. A brief discussion is added in the main text (Page 5). A detailed discussion is provided in Supporting Information (Page 5).

(2) We thank the reviewer for his/her good suggestion about the XAS measurement. We measured the O K-edge X-ray absorption spectra of 3C-SrIrO₃ and 6H-SrIrO₃ (see Figure S13 in SI). The result reveals that there is a stronger Ir 5d-O 2p hybridization or stronger Ir-O covalence for 3C-SrIrO₃. This is contrary to the result expected by the reviewer. However, our result, coupled with the recent studies by Shao-Horn *et al.* (see reference: Perovskites in catalysis and electrocatalysis. *Science* **358**, 751–756 (2017)), might provide a good explanation why 6H-SrIrO₃ has a better structural stability than 3C-SrIrO₃. The following discussion has been added in the revised manuscript (see page 12). A detailed discussion about Figure S13 is provided in Supporting Information (Page 20).

“...The better structural stability of 6H-SrIrO₃ in comparison with 3C-SrIrO₃ can be explained by the comparison of their O K-edge X-ray absorption spectra (Figure S13, SI). The comparison reveals that there is stronger Ir 5d-O 2p hybridization or stronger Ir-O covalence for 3C-SrIrO₃. The stronger covalence results in the easier structural loss of the perovskite phase or surface amorphourization during electrocatalysis, as suggested by the recent studies on perovskite electrocatalysts...”

Comment 3: I am surprised to see atomic distances in Figure 1 while in Figure 2 no Rietveld analysis was performed on the XRD obtained after synthesis. The authors must perform such analysis in order to discuss the atomic distances and make sure that the material they prepare is indeed 6H SrIrO₃ with similar crystallographic properties than the ones reported in the literature. Same applies to the electrical conductivity that the authors did not measure in this work but rather infer from previous works. Caution must be exercised since slight differences in oxygen content or atomic arrangements due to the synthesis protocol can drastically modify these properties.

Response 3: Thanks!

(1) We performed the Rietveld analysis on the powder X-ray diffraction pattern of the material. The result has been provided in Figure 2a. The observed powder X-ray diffraction pattern is almost in agreement with the fitting plot (Figure 2a), revealing that the material is composed of high purity 6H-SrIrO₃. In addition, our result is also similar with those reported previously (*e.g.*, J. Solid State Chem. 1971, 3, 174; J. Mater. Chem. A, 2013, 1, 3127).

(2) We measured the electrical conductivity (or the electrical resistivity) of 6H-SrIrO₃, and the result was provided in Figure S3 in Supporting Information. Correspondingly, the description about the electrical resistance of 6H-SrIrO₃ was also modified in the revised manuscript. The modified description is also provided here (see below).

“...This is further supported by the experimental result (Figure S3 in SI). The electrical resistivity of 6H-SrIrO₃ was measured to be $\sim 2.7 \times 10^{-4} \Omega \cdot \text{m}$ at room temperature...”

Comment 4: Concerning the chemical stability of the phase in 0.5M H₂SO₄, when the authors claim that “1.3% of the Ir is leached out under these conditions”, this value is not negligible. In

fact, it would be interesting to compare this value with the one obtained for IrO₂. Indeed, it seems very high especially when compared to value often measured for IrO₂ (see work from Mayrhofer and others for comparison). Same applies for Ir leaching during OER, it should be compared with IrO₂ or the value obtained should be discussed with what was previously reported for IrO₂. Only providing this comparison will allow for stating that 6H SrIrO₃ is indeed more stable than other catalysts.

Response 4: Thanks!

(1) We thought that the reviewer could misunderstand our experimental result about the chemical stability of 6H-SrIrO₃. Our result shows that after such a long time testing (48 days), there is **no detectable leached Ir species** and **only ~1.3% of total Sr content** leached in the acidic solution (the same discussion is also shown in Pages 7 and 8 in the main text).

(2) As suggested by the reviewer, we compared the Ir leaching during OER in the presence of 6H-SrIrO₃ and IrO₂. We used inductively coupled plasma atomic emission spectroscopy (ICP-OES) to detect the Ir species in the electrolyte at a current density of 10 mA/cm² with 6H-SrIrO₃ or IrO₂ as the electrocatalyst. The result (Figure S14, in SI) reveals that a tiny amount of leached Ir species are detected in the electrolytes for both 6H-SrIrO₃ and IrO₂ as the electrocatalysts, but the amount of leached Ir species in the presence of 6H-SrIrO₃ as the electrocatalyst is obviously lower than that in the presence of IrO₂ as the electrocatalyst. This result further demonstrates that 6H-SrIrO₃ is a stable electrocatalyst for OER in acid.

The following discussion has been added in the revised manuscript.

“...In order to further assess the stability of 6H-SrIrO₃, we compared the Ir leaching during OER in the presence of 6H-SrIrO₃ and IrO₂. We used inductively coupled plasma atomic emission spectroscopy (ICP-OES) to detect the Ir species in the electrolyte at a current density of 10 mA/cm² with 6H-SrIrO₃ or IrO₂ as the electrocatalyst. The result (Figure S, in SI) reveals that a tiny amount of leached Ir species are detected in the electrolytes for both 6H-SrIrO₃ and IrO₂ as the electrocatalysts, but the amount of leached Ir species in the presence of 6H-SrIrO₃ as the electrocatalyst is obviously lower than that in the presence of IrO₂ as the electrocatalyst. This result further demonstrates that 6H-SrIrO₃ is a stable electrocatalyst for OER in acid...”

Comment 5: Concerning the OER activity, the authors should avoid statement such as “the most active Ir-based oxide electrocatalysts for OER in acid”; page 9. Indeed, protocols for measuring the different phases might be different (such as loading, use of binder or others). Same applies for the statement concerning the mass activity.

Response 5: Thanks for the reviewer’s comments herein. We have modified the corresponding statements (please see page 9 in the main text). The modified sentence is: “The results also reveal that 6H-SrIrO₃ is among the most active Ir-based oxide electrocatalysts for OER in acid (see Table S2, SI), although there might be the differences in protocols, such as catalyst loading, use of binder and electrolyte, employed in their electrocatalytic tests.”

Comment 6: Page 10, when describing the “as-synthesized” cubic SrIrO₃ that has a deficiency of Sr due to acid leaching what the authors mean by as-synthesized? The EDX has been taken after synthesis, or after acid treatment mimicking the one performed by Seitz et al? That should be clarified.

Response 6: Thanks for the reviewer’s good questions. We also want to say sorry for the unclear description about the synthetic procedures of SrIrO₃ phases and the reasons for the acid treatment during the synthesis.

First, in order to synthesize the SrIrO₃ phases (both 6H-SrIrO₃ and 3C-SrIrO₃) without IrO₂ as the secondary phase, an excessive amount of strontium (Sr:Ir atomic ratio = 8:1) was used during the synthesis in this work. The use of the excessive amount of strontium resulted in the formation of strontium carbonates (SrCO₃) as the impurities (please see Figure S18 in Supporting Information). The acid treatment with 1M HCl was employed to remove the SrCO₃ impurities.

The corresponding discussion was also involved in the revised manuscript (see pages 17 and 18).

Second, the “as-synthesized” cubic SrIrO₃ is the material obtained after 1M HCl treatment (or the material after the removal of SrCO₃ impurities). Additionally, the EDX was performed on the 3C-SrIrO₃ material after the synthesis (*i.e.*, the material after the removal of SrCO₃ impurities *via* acid treatment). The following discussion has been added in the revised manuscript (see page 11).

“...The 3C-SrIrO₃ material has a Sr:Ir atomic ratio of 0.82:1 and a certain amount of IrO_x on the surface (Figure S10, SI). The Sr deficiency for 3C-SrIrO₃ is because the acid treatment is necessary for the removal of the SrCO₃ impurities during the material synthesis, and simultaneously, the acid treatment also results in some strontium leached from 3C-SrIrO₃...”

Comment 7: When discussing the thermodynamical potential obtained by DFT, the authors do not discuss why the overpotential computed by DFT is by far greater than the one experimentally obtained (almost 200 mV greater). Where this discrepancy comes from? This should be discussed in greater details. Moreover, in addition to compare the computed overpotential depending on the surface studied for 6H SrIrO₃, authors should compare directly with the cubic SrIrO₃. Indeed, even if the oxygen atoms are coordinated in a corner-shared fashion in surface II, their energy will be dependent on the subjacent structure.

Response 7: Thanks! We added the discussion about the difference between the thermodynamic overpotentials obtained by DFT and the experimental ones in the revised manuscript (see Page 16 in the main text or see below).

“...When DFT calculations have provided some useful insights into the catalytic active sites and the activity trend for the Ir-based materials from the theoretical computations is consistent with that from our experiments, there is difference between the thermodynamic overpotentials obtained by DFT and the experimental ones. This difference might be caused by multiple factors.

First, all the computations are based on thermodynamic considerations. Other important electrocatalytic processes, such as mass and charge transport, can not be involved in the DFT calculations. Second, the distribution and concentration of active sites on the real material surface are generally very complex, but this complexity can not be adequately reflected by DFT calculations. Third, at current stage, DFT calculations are not capable of providing atomic information on the more complex catalyst-electrolyte interface, which is crucially important for the electrocatalysis process...”

Moreover, as suggested by the reviewer, we performed the parallel computations to investigate the OER catalytic activity of cubic SrIrO₃ (*i.e.*, 3C-SrIrO₃). The result has been provided in Figure S 17, SI.

Comment 8: Page 15, when the authors explain that their DFT study is consistent with their “analysis” of the Ir-O bond being weaker for the face-shared octahedral, this claim is not supported. Indeed, rather than an “analysis”, the authors only discuss in the first part of their manuscript very general feelings that are not supported by any measurements. Moreover, when discussing about the nucleophilic attack of water on the oxygen sites, this mechanism has been discussed by Strasser and Grimaud as resulting from the oxidation of surface oxygen under OER conditions, which eventually can result into degradation and instability of the materials. Why would that be different for 6H SrIrO₃? Rather than this chemical acid-base step, it seems that the 6H perovskite behaves following the classical PCET mechanism.

Response 8: Thanks for the reviewer’s valuable comments.

--- I agree the reviewer’s comment on our discussion about the relationship between DFT study and “analysis” of the bulk structure. We also think that our discussion about this relationship is farfetched. Thus, we deleted the concerned discussion from the main text. The deleted discussion is also shown as bellow.

This discussion has been deleted from the main text: “This result is also in agreement with our above analyses of the 6H-SrIrO₃ crystal structure, where the Ir-O bonds in the face-sharing IrO₆ octahedral dimers (the main structural feature of surface-I) are significantly weaker than those in the isolated, corner-sharing IrO₆ octahedra (the main structural feature of surface-II).”

---I agree the reviewer’s comment on our discussion about the nucleophilic attack of water on the oxygen sites. In fact, we misunderstood the mechanism discussed by Strasser *et al.* Initially, we would like to speculate why the surface-I had better catalytic activity. Obviously, it is not correct. So, we deleted the concerned discussion from the main text. The deleted discussion is also shown as bellow.

This discussion has been deleted from the main text: “Furthermore, it is speculated that the relatively weak surface Ir-O bonds and the corresponding low electronic density of O atoms in them for the surface-I should be conducive for the following nucleophilic attack of water

molecules for the formation of HOO intermediates, that is, for accelerating the potential-determining reaction step involved in the OER (*i.e.*, $O^* + H_2O \rightarrow HOO^* + H^+ + e^-$.)”

Comment 9: Finally, the methods section is lacking a lot of details, such as the characterizations techniques or purity of precursors or acids etc. This must be fixed in order to allow for good reproducibility of these results.

Response 9: We provided the details about the chemicals and reagents as well as the characterizations techniques in the Supporting Information (Please see Page 2 in the Supporting Information).

Comment 10: Additional comments:

Page 9: La_2LiIrO_6 is a perovskite, not a pyrochlore

Numerous typos such as actahedral throughout the text, “structure” page 5, potanitals page 13 etc.

Microsized should be micronsized page 10

Page 9, when referring to the mass activity, it should not be Figure 2b but rather Fig. 3b.

Response 10: Many thanks! Those mistakes have been corrected in the revised manuscript.

Reviewer: 2

Comment 1: Yang and coworkers report the OER on 6H-SrIrO₃ and found the OER to be more active than 3C-SrIrO₃, IrO₂, and Iridium-oxide pyrochlores. The search for high-activity OER catalysts is an important direction in energy science. The high OER performance of 6H-SrIrO₃ is therefore a welcome result. However, there are a few technical concerns, which should be fully and thoroughly addressed.

Response 1: We thank the reviewer’s comments here! We have addressed the comments point-by-point as follows.

Comment 2: Synthesis. Why do the authors choose to use a series of heat treatment (200 °C for 6h, 300 °C for 6h, 500 °C for 3h, and 700 °C for 6h) for the 6H-SrIrO₃ synthesis? Was this synthesis randomly discovered or was there a rationale behind it?

Response 2: Thanks! Such a temperature-programming was determined based on our many experimental attempts. Our experimental results show that such a temperature-programming can ensure that the synthesis is highly reproducible/repeatable.

Comment 3: 6H-SrIrO₃ is a stable phase at RT, whereas 3C-SrIrO₃ is not. Why does adding more citric acid favor the metastable 3C-SrIrO₃ formation over 6H-SrIrO₃?

Response 3: 6H-SrIrO₃ is, indeed, the thermodynamically stable phase, in comparison with 3C-SrIrO₃ (please see literature: *e.g.*, *J. Solid State Chem.* 2016, 238, 74-82). This argument can also be supported by our calculation result, which shows that the formation energy (-8.26 eV) of 6H-SrIrO₃ is a little lower than that (-8.13 eV) of 3C-SrIrO₃ (see calculation method in the literature: *Angew. Chem. Int. Ed.* 2010, 49, 7699). Our calculation result further suggests that although 6H-SrIrO₃ is thermodynamically more stable, the stability difference between 6H-SrIrO₃ and 3C-SrIrO₃ is not large. This should be the thermodynamic reason why the formation of the metastable 3C-SrIrO₃ is possible by the fine tuning of synthetic system (*e.g.*, the addition of more citric acid in our synthesis). Nevertheless, to be honest, the atomic/molecular basis for the synthesis of the metastable 3C-SrIrO₃ is still unclear at current stage, and calls for further in-depth mechanistic study.

Comment 4: Why are the BET areas of 6H-SrIrO₃ and 3C-SrIrO₃ so different from IrO₂ when they were both synthesized using the same method, simply without adding Sr?

Response 4: 6H-SrIrO₃ and 3C-SrIrO₃ have much lower BET areas in comparison with IrO₂. This result is in agreement with the fact that 6H-SrIrO₃ and 3C-SrIrO₃ have micron-sized particles, whereas IrO₂ is composed of nanosized particles. In view of the similar synthetic environment for both SrIrO₃ materials and IrO₂, their different BET areas and particle sizes might be mainly originated from their different crystal growth habits. It is inferred that the nucleation of IrO₂ should be easier compared with 6H-SrIrO₃ and 3C-SrIrO₃ and thus a larger amount of crystal nucleus should be generated during the synthesis of IrO₂, resulting in the growth of nanosized IrO₂ as the final product.

Comment 5: What impurities are the authors getting rid of with the 1M HCl treatment?

Response 5: Thanks! In order to synthesize the SrIrO₃ phases without IrO₂ as the secondary phase, an excessive amount of strontium (Sr:Ir atomic ratio = 8:1) was used during the synthesis in our work. The use of the excessive amount of strontium resulted in the formation of strontium carbonates (SrCO₃) as the impurities (please see Figure S18 in Supporting Information). The acid treatment with 1M HCl was employed to remove the SrCO₃ impurities.

The corresponding discussion was also involved in the revised manuscript (see pages 17 and 18).

Comment 6: The authors argue that the surface structure of 6H-SrIrO₃ is stable because very little Sr leaches out after the OER testing. There are three questions regarding this claim.

a. Could Sr leach out during the 1M HCl treatment? If so, would not the overall Sr loss be greater than the amount of Sr measured in the electrolyte?

-
- b. ~1% Sr dissolution is significant for micron-sized particles and can correspond to a few monolayer dissolutions. The authors should estimate how many monolayers this is.
- c. It is difficult to use TEM to quantify the first few surface monolayers. Supplemental methods should be used to verify that the surface structure of 6H-SrIrO₃ is stable.

Response 6: We thank the reviewer for these good questions.

Our response to the question a: No, Sr species are not leached during the 1M HCl treatment.

Our response to the question b: We estimated about 0.9-1.5 layers of Sr for 6H-SrIrO₃ dissolved during electrocatalysis (see Page 11).

Our response to the question c: We provided the comparative result on XPS spectra of the materials before and after electrocatalysis (see Figure S12, SI). The result further supports the structural stability of 6H-SrIrO₃. The discussion on the XPS spectra is also provided in the Supporting Information (or see below).

“X-ray photoelectron spectroscopy (XPS) was employed to study the local structure of Sr species for 6H-SrIrO₃ and 3C-SrIrO₃ before and after 30 hours of electrocatalysis testing (Figure S12). The Sr3d XPS spectrum of Sr-containing perovskite is well known to be sensitive to the surface structural rearrangement of the material. As shown in Figure S12a and S12c, the XPS spectra of 6H-SrIrO₃ and 3C-SrIrO₃ before the electrocatalysis are similar, and their Sr3d peaks can be fitted with two doublets. The first doublet, shown in black, is assigned to the lattice Sr (or bulk Sr), and the second doublet, shown in grey, is attributed to the surface Sr. Comparison of the Sr3d XPS spectra of 6H-SrIrO₃ (Figure S12a and S12b) shows that the first doublet keep unchanged, and the second doublet decrease to a certain extent. This result demonstrates that 6H-SrIrO₃ can keep its bulk structure intact during electrocatalysis, with a small amount of surface Sr leaching. Comparison of the Sr3d XPS spectra of 3C-SrIrO₃ (Figure S12c and S12d) shows that the first doublet almost disappears, and the second doublet decreases significantly. This result, in agreement with other characterization results (Figure 3), reveals that 3C-SrIrO₃ loses a large amount of Sr and undergoes surface amorphization during electrocatalysis.”

Comment 7: Page 7, the statement that “the exposed facets of micron-sized 6H-SrIrO₃ plates is {001}” should be supported with orientation mapping data from EBSD.

Response 7: Thanks for the reviewer’s good suggestion. We have attempted my times to obtain the orientation mapping data from EBSD, but all my attempts failed. The reason for the failure lies in the small thickness (< 100 nm) of 6H-SrIrO₃ plates. Because the detection depth of EBSD is obviously larger than the thickness of the sample, it becomes impossible to get the useful orientation mapping data without the interference of the base material (*i.e.*, Si plate in our experiment). Actually, the sample signals are often buried in the signals from the base material. So, we have to say sorry for not providing the orientation mapping data from EBSD.

Here, we would like to add that although EBSD is a mature characterization method, but it is rarely used for the analysis of growth orientation of powdered metal oxides. By contrast, TEM (or HRTEM) is a very common, reliable technique to determine the exposed facets of powdered metal oxides. That is why we use this technique to study the exposed facets of 6H-SrIrO₃ in our work. Figure 2c presents a high-quality HRTEM image of 6H-SrIrO₃, in which the interplanar distances and the angles between the crystallographic planes can be obtained. These data obtained from the HRTEM image can further be used for the determination of the exposed facet.

We hope that we have addressed the comment raised by the reviewer.

Comment 8: Was the reference calibration done using hydrogen redox tested in the same electrolyte? If not, this should be done since small ionic activities inside the reference can shift its true potential.

Response 8: The reference calibration was done in the same electrolyte (*i.e.*, 0.5 M H₂SO₄). The details for the calibration of the reference electrode were provided in Supporting Information (see Page 2-3 in Supporting Information).

Comment 9: The authors should compare the DFT theoretical overpotential for IrO₂, 6H-SrIrO₃, and 3C-SrIrO₃ in the same figure (Ref. 30 and 31). The authors should also calculate IrO₂ to ensure that their DFT models are consistent with all three materials in order to ensure fair comparisons.

Response 9: As suggested by the reviewer, we performed the parallel computations to investigate the OER catalytic activities of IrO₂ and 3C-SrIrO₃. The result has been provided in Figure S. The corresponding discussion has been added in the main text.

Reviewer: 3

Comment 1: Review on “Efficient Oxygen Evolution Electrocatalysis in Acid by an Unusual Perovskite Oxide with Face-Sharing IrO₆ Octahedral Dimers”.

The lack of good alternative to Ir based OER catalysts in acidic environments is a huge challenge for electrocatalysis today. The approach in this study is to reduce the amount and increase the activity of Ir, which is a valid approach. The issue of Ir for OER is similar to Pt for ORR, except that OER probably is more important than ORR and Ir is rarer than Pt. Whereas, several different Pt based catalysts for ORR are presented in top journals every year, Ir based OER catalysis is much less studied. Therefore, I find the subject of this paper highly relevant and interesting for Nature Comm.

Response 1: We appreciate the reviewer’s positive comments about our work.

Comment 2: The activities compared to IrO₂ is per geometric surface area and per mass. The activity per electrochemical surface area is discussed in text. The conclusion is that on all measures the perovskite outperforms IrO₂. This suggests that at least part of the increased activity is related to more active sites on perovskite surface (which the authors also write). This leads to DFT simulations, where the analysis follows ref. 31. In the overpotential the difference is ~0.04 V (measured/geo area). The DFT simulations of the free energy diagrams support the difference between two surfaces, but IrO₂ *e.g.* (110) was not included. I think free energy diagram of pure IrO₂ should be included in fig 4 for reference.

Response 2: According to the reviewer's comment, we have performed the computations to examine the OER catalytic activity on the sampled (110) surface for IrO₂. The computed overpotential value for IrO₂ is 0.59 V, which is comparable to that of the previously reported one (0.56 V) (Journal of Electroanalytical Chemistry 2007, 607, 83). The result has been provided in the Figure S 16, in SI.

Comment 3: The reason surface I is found more active than II is due to the weaker oxygen binding and due to the almost constant OOH binding. The difference between surface I and II, does not exactly follow the scaling between OH and OOH. Therefore, O and OH could be in local minima structures, I would like the authors to test that the O and OH structures of surface I cannot be relaxed to slightly more stable structures. This could be done by starting in the OOH structure and remove OH.

Response 3: Thanks! As suggested by the reviewer, we have recomputed the O and OH structures related to the surface-I and surface-II by starting from the corresponding OOH structure. Our computed results show a slight change for some O and OH structures, and the resulting overpotential values can be almost maintained. All these results have been updated in the revised version. Additionally, it is noteworthy that the present free energy difference between HO* and HOO* is 3.01 eV for surface-I and 3.15 eV for surface-II, respectively, both of which are in the scope of the scaling relationship within 3.2 ± 0.2 eV (see reference: *e.g.*, ChemCatChem 2011, 3, 1159)

Reviewers' comments:

Reviewer #1 (Remarks to the Author):

The authors successfully responded to all my comments, and the manuscript can now be accepted.

Reviewer #2 (Remarks to the Author):

Comments from Reviewer 2:

I do not find the response to be satisfactory. The authors should directly respond to the questions instead of repeating the questions or making claims that are unsupported. I cannot recommend the work to be accepted until all the concerns are fully and entirely addressed.

Comment 2: The authors did not answer the question. Was the synthesis condition randomly discovered or was there a rationale behind it?

Comment 3: The authors did not answer the question. Why does adding more critic acid favor the formation of the metastable 3C-SrIrO₃ phase over 6H-SrIrO₃?

Comment 4: The authors claimed that IrO₂ nucleated faster than 3C-SrIrO₃ and 6H-SrIrO₃. What evidence do the authors have that supports this claim? What if instead the IrO₂ growth was arrested?

Comment 6a: What evidence do authors have that suggest that Sr did not dissolve during the 1M HCl heat treatment? This information should be provided in the manuscript.

Comment 6b: Given that 0.9-1.5 layers of Sr in 6H-SrIrO₃ were lost during the OER, should not the OER active layer be represented as IrO_x/6H-SrIrO₃, similar to that of IrO_x/3C-SrIrO₃, as opposed to 6H-SrIrO₃? Also, the authors should show the isotherm data for both 6H-SrIrO₃, IrO₂, and 3C-SrIrO₃ before and after the OER to show that no micropore was formed after Sr was lost.

Comment 6C: Since the XPS penetration depth (~2 nm) is larger than the IrO_x layer (~0.9-1.5 layers), the XPS signal is likely dominated by 6H-SrIrO₃ underneath. Therefore, it is not surprising that the XPS is the same before and after the OER. Instead, other methods to test the surface quality should be used. For example, the authors may compare the CV between 0-1.2 V vs. RHE on the first scan and after a few hours for 6H-SrIrO₃ vs. IrO₂ vs. 3C-SrIrO₃ to show whether they are the same or different. The CV result can also help support the authors' claim that 6H-SrIrO₃ has a much smaller surface area than IrO₂.

Comment 7: I have questions with the authors' comment that EBSD cannot be conducted because 6H-SrIrO₃ was thin (< 100 nm). First, what evidence do the authors have that 6H-SrIrO₃ was < 100 nm thick? If it is via TEM, the authors should report the particle size distribution for all x-, y-, and z-axes to show that the majority of the thickness is < 100 nm. Second, the EBSD depth is 50-100 nm, so it is indifferent to whether the thickness of the sample is > 100 nm or not (Nowell et al. *Microscopy Today*, 13, 2005). Instead, the authors should consider the possibility that the EBSD signal was not observed because the surface was amorphous (Small et al. *J. Microscopy*, 206, 2002).

Comment 9: On the models used for DFT, since the active layer for the OER is IrO_x/6H-SrIrO₃ for 6H-SrIrO₃ and IrO_x/3C-SrIrO₃ for 3C-SrIrO₃, should not these structures be compared for the DFT calculation? What is the supporting evidence behind the used 6H-SrIrO₃ structure? If there is

supporting evidence for the used 6H-SrIrO₃ structure, did the authors remove that the oxygen atoms together with the Sr removal to keep the charge neutral? If there is no evidence, why don't the authors use the structure proposed in Ref. 16 with 6H-SrIrO₃ as the underlayer?

Finally, there are several grammatical errors in the work. The authors should consider consulting a professional proofreader.

Reviewer #3 (Remarks to the Author):

The authors have answered all my questions as suggested. Therefore, I see no reason why it should not be accepted and I recommend that the study is published in Nature Comm

Reviewer #2's Comments and Our Responses:

Reviewer: 2

Comment: I do not find the response to be satisfactory. The authors should directly respond to the questions instead of repeating the questions or making claims that are unsupported. I cannot recommend the work to be accepted until all the concerns are fully and entirely addressed.

Response: We appreciate the further comments of Reviewer #2, which have definitely enabled us to improve the manuscript. We have addressed the comments point-by-point as follows.

Comment 2: The authors did not answer the question. Was the synthesis condition randomly discovered or was there a rationale behind it?

Response 2: The synthesis condition was randomly discovered.

Comment 3: The authors did not answer the question. Why does adding more critic acid favor the formation of the metastable 3C-SrIrO₃ phase over 6H-SrIrO₃?

Response 3: Thanks! In order to address the reviewer's question, some experiments were performed, and the following discussion is added in the revised manuscript.

“...In order to explore the reasons why the addition of more critic acid in the synthesis can favor the formation of metastable 3C-SrIrO₃ over 6H-SrIrO₃, we first compared the Ir4f XPS spectra of IrO₂, 6H-SrIrO₃ and 3C-SrIrO₃ (Figure S11a, SI). The results reveal that the Ir 4f_{5/2} and Ir 4f_{7/2} XPS peaks for 3C-SrIrO₃ appear at the higher binding energies compared with IrO₂ and 6H-SrIrO₃. This demonstrates that the average oxidation state of iridium for 3C-SrIrO₃ is higher than 4+.³⁰ The higher average oxidation state of iridium are believed to be favorable for the formation of 3C-SrIrO₃. This argument is mainly based on recent studies, in which low-valent heteroatom doping (*e.g.*, Co, Mg, Zn, *etc.*) can lead to the increase in the oxidation state of iridium and the stablization of 3C-SrIrO₃.^{28,31,32} In view of the difference in synthetic systems between our work and the previous studies,^{28,31,32} we attempted to synthesize 3C- phase SrIrO₃ by low-valent heteroatom doping (cobalt as dopants used in this work) with the experimental procedures that were used for the synthesis of 6H-SrIrO₃ (see experimental details in Methods). The results (Figure S11b, SI) show that in this case, the formation of the 3C-phase is indeed preferable, and the resulting material also has a higher oxidation state of iridium. Next, we compared the thermogravimetric (TG) and differential thermal analysis (DTA) in air of the precursors for the synthesis of 6H-SrIrO₃ and 3C-SrIrO₃. As shown in Figure S12, SI, the organic components in the precursors are oxiditively decomposed in air at elevated temperature, and this process is highly exothermic. The presence of more critic acid (*i.e.*, the precursor for the

synthesis of 3C-SrIrO₃) is also found to result in more heat released at elevated temperature. The more released heat in the synthesis should favor the generation of high-valent iridium species and thereby the formation of 3C-phase SrIrO₃...”

Comment 4: The authors claimed that IrO₂ nucleated faster than 3C-SrIrO₃ and 6H-SrIrO₃. What evidence do the authors have that supports this claim? What if instead the IrO₂ growth was arrested?

Response 4: Thanks for the reviewer’s questions. We also want to say sorry for not providing a satisfied response to the reviewer’s question in the first-round revision.

Additionally, we also think that in our first-round response, the explanation on the larger surface area of IrO₂ by its faster nucleation is not reasonable. And the faster IrO₂ nucleation can also not be supported by any experimental results. Thus, we try my best to re-reply the reviewer’s comment herein. We hope that we can address the question raised by the reviewer.

The comment in the first round revision is coped herein.

Original Comment 4: Why are the BET areas of 6H-SrIrO₃ and 3C-SrIrO₃ so different from IrO₂ when they were both synthesized using the same method, simply without adding Sr?

Our new response 4: The larger BET surface area of IrO₂ in comparison with 6H-SrIrO₃ and 3C-SrIrO₃ (Table S2 in SI) is because the former has smaller particle size. The distribution of particle size for these materials have been provided in the Supporting Information (see Figure S4 S8 and S13, SI).

Some important, relevant discussion is also provided in the revised manuscript (or see below). “...Scanning electron microscopy (SEM) image (Figure 2a and S4, SI) shows that the material has micron-sized, plate-like particles with a dominant thickness distribution of 30-60 nm.” (**Page 7**)

“...we also synthesized IrO₂ nanoparticles with a particle size of 10-20 nm (see structural characterizations in Figure S8, SI)” (**Page 9**)

“...the IrO₂ sample has a larger BET surface area (19.8 m²/g) than 6H-SrIrO₃, due to the former’s smaller particle size (see Figures S4 and S8, SI). This is in agreement with the effective electrochemical active surface area of IrO₂, which is measured to be about 3 times larger than that of 6H-SrIrO₃ (Figure S9, SI)...” (**see page 10**)

Comment 6a: What evidence do authors have that suggest that Sr did not dissolve during the 1M HCl heat treatment? This information should be provided in the manuscript.

Response 6a: Thanks! First, we want to clarify a fact that 1M HCl treatment, rather than 1M HCl *heat* treatment, was used for the materials synthesis in our work.

Our evidences to support that Sr did not dissolve during 1M HCl treatment mainly include:

(1) The Sr/Ir mol ratio is *ca.* 1:1 for 6H-SrIrO₃, as revealed by the chemical mapping/energy dispersive X-ray spectroscopy (EDS) and X-ray photoelectron spectroscopy (XPS) results. The following discussion has been added in the revised manuscript (see page 7).

“...Additionally, chemical mapping and energy dispersive X-ray spectroscopy (EDS) show that the Sr and Ir elements are homogeneously distributed over the material, with a Sr:Ir atomic ratio of *ca.* 1:1 (Figure S4 in SI). X-ray photoelectron spectroscopy (XPS) result further reveals a Sr:Ir atomic ratio of *ca.* 1:1 for 6H-SrIrO₃...”

(2) In order to further assess the stability of 6H-SrIrO₃ in 1M HCl solution, we dispersed a certain amount of 6H-SrIrO₃ powders in 1M HCl solution for two days, and then used ICP-OES to detect the Sr and Ir ions in the solution. The result shows that there is no detectable leached Sr and Ir species in the solution, demonstrating the good stability of 6H-SrIrO₃ in 1M HCl solution.

This experiment and the corresponding result have been provided in the revised manuscript (see page 19).

Comment 6b: Given that 0.9-1.5 layers of Sr in 6H-SrIrO₃ were lost during the OER, should not the OER active layer be represented as IrO_x/6H-SrIrO₃, similar to that of IrO_x/3C-SrIrO₃, as opposed to 6H-SrIrO₃? Also, the authors should show the isotherm data for both 6H-SrIrO₃, IrO₂, and 3C-SrIrO₃ before and after the OER to show that no micropore was formed after Sr was lost.

Response 6b: Our results reveal that there are 0.9-1.5 layers of Sr leached in 6H-SrIrO₃ during the OER, but there was not secondary, amorphous IrO_x phase generated on the 6H-SrIrO₃ surface. In other words, 6H-SrIrO₃ had a Sr-deficient surface during the OER. Thus, the 6H-SrIrO₃ catalyst is not suitable to be represented as IrO_x/6H-SrIrO₃.

The isotherm data for 6H-SrIrO₃, IrO₂, and 3C-SrIrO₃ before the OER are provided in Figures S8 and S13, SI. The isotherm data for 6H-SrIrO₃, IrO₂, and 3C-SrIrO₃ after the OER are provided in Figures S17, SI. The results show that there are not micropores for these three materials before and after OER.

Comment 6c: Since the XPS penetration depth (~2 nm) is larger than the IrO_x layer (~0.9-1.5 layers), the XPS signal is likely dominated by 6H-SrIrO₃ underneath. Therefore, it is not surprising that the XPS is the same before and after the OER. Instead, other methods to test the surface quality should be used. For example, the authors may compare the CV between 0-1.2 V vs. RHE on the first scan and after a few hours for 6H-SrIrO₃ vs. IrO₂ vs. 3C-SrIrO₃ to show whether they are the same or different. The CV result can also help support the authors' claim that 6H-SrIrO₃ has a much smaller surface area than IrO₂.

Response 6c: Thanks for the reviewer's comment on the XPS characterization.

As suggested by the reviewer, we compared the CV curves of the materials before and after OER (see Figure S17, SI). The comparison of the CV curves reveals that there was not secondary, amorphous IrO_x phase generated on the 6H-SrIrO₃ surface.

In addition, based on the CV results, the effective electrochemical active surface areas of the materials were also measured. The result shows that the effective electrochemical active surface area of 6H-SrIrO₃ is about 3 times lower than that of IrO₂ (Figure S9, SI).

Comment 7: I have questions with the authors' comment that EBSD cannot be conducted because 6H-SrIrO₃ was thin (< 100 nm). First, what evidence do the authors have that 6H-SrIrO₃ was < 100 nm thick? If it is *via* TEM, the authors should report the particle size distribution for all x-, y-, and z-axes to show that the majority of the thickness is < 100 nm. Second, the EBSD depth is 50-100 nm, so it is indifferent to whether the thickness of the sample is > 100 nm or not (Nowell *et al.* Microscopy Today, 13, 2005). Instead, the authors should consider the possibility that the EBSD signal was not observed because the surface was amorphous (Small *et al.* J. Microscopy, 206, 2002).

Response 7: Thanks for the reviewer's comment.

The thickness of 6H-SrIrO₃ was determined *via* SEM. As suggested by the reviewer, we provide the particle size distribution in the Figure S4, SI. The result shows that the material has micron-sized, plate-like particles with a dominant thickness distribution of 30-60 nm. This particle thickness is around the lower limit of EBSD detection.

We agree with the reviewer's point that the EBSD signal can be influenced by the surface amorphous phase. However, 6H-SrIrO₃ is a highly crystalline material without amorphous phase on its surface. This can be supported by our high-resolution TEM and the fast Fourier transform results (see Figure 2c) as well as the CV results (Figure S17, SI).

Comment 9: On the models used for DFT, since the active layer for the OER is $\text{IrO}_x/6\text{H-SrIrO}_3$ for 6H-SrIrO₃ and $\text{IrO}_x/3\text{C-SrIrO}_3$ for 3C-SrIrO₃, should not these structures be compared for the DFT calculation? What is the supporting evidence behind the used 6H-SrIrO₃ structure? If there is supporting evidence for the used 6H-SrIrO₃ structure, did the authors remove that the oxygen atoms together with the Sr removal to keep the charge neutral? If there is no evidence, why don't the authors use the structure proposed in Ref. 16 with 6H-SrIrO₃ as the underlayer?

Response 9: Thanks for the reviewer's valuable comments.

Our results reveal that there are 0.9-1.5 layers of Sr leached in 6H-SrIrO₃ during the OER, but there are not secondary, amorphous IrO_x phase generated on the 6H-SrIrO₃ surface. In other words, 6H-SrIrO₃ has a Sr-deficient surface. Thus, we constructed the model of 6H-SrIrO₃ for DFT calculations by removing the Sr atoms on the surface together with the oxygen atoms to keep the charge neutral (see details in Supporting Information, Page 4 and 5).

Comment: Finally, there are several grammatical errors in the work. The authors should consider consulting a professional proofreader.

Response: Thanks for the reviewer's suggestion. We have tried our best to improve the English writing.

We hope that we have addressed the comments and suggestions raised by the reviewers. We look forward to hearing your decision at your convenience please. Thanks a lot!

With best regards,

Sincerely yours,

Xiaoxin Zou

Reviewers' comments:

Reviewer #2 (Remarks to the Author):

The authors have not presented sufficient evidence to conclusively show that the surface of 6H-SrIrO₃ is not amorphous. If the authors can provide the evidence to support this claim, most of the concerns will be addressed. The authors have so far presented HRTEM, which cannot be used to resolve the topmost atomic layer of nanomaterials, and CV, which show that the surface chemistry changes before and after the OER. The authors argued based on the comparison between the CV of 6H-SrIrO₃ and IrO_x that the surface of 6H-SrIrO₃ is not amorphous. However, as there are many types of amorphous IrO_x (which is why some amorphous IrO_x are more active than the others, for example, in Ref. 16), one comparison is not sufficient and cannot be used to conclude that the surface is crystalline. For this reason and other weaknesses in the lack of information on the synthesis side, I cannot recommend an acceptance of the work until all the concerns are fully and entirely addressed.

Comment 2: The authors' response is satisfactory. I suggest that the authors include a claim that the sequence of synthesis steps was refined through trials and errors (or other wordings of that nature) in the revision. This will help clearly indicate that there is no specific reason why the authors took several steps to make 6H-SrIrO₃, which due to its complex series of steps can be confusing to the readers.

Comment 3: The extra experiments are appreciated. However, the response is not satisfactory. The authors are indirectly suggesting that the presence of a citric acid favors the formation of 3C-SrIrO₃ by stabilizing the more oxidized form of iridates. First, the authors should directly state this claim instead of coyly saying "some experiments were performed". Second, the citric acid is a reducing agent. Should not it drive iridium toward a more reduced state, which would consequently destabilize 3C-SrIrO₃?

Comment 4: The response is not satisfactory. It is a common understanding that smaller particles have higher surface areas. Stating the obvious fact that materials with higher surface areas have smaller sizes is not helpful and does not address the original question of why the areas/sizes are different.

Comment 6a: The authors' ICP-OES experiment satisfactorily addresses the concern.

Comment 6b: The authors' response is not satisfactory. There are several concerns. First, the authors claim that there is no amorphous IrO_x layer. Please provide experimental evidence to support this claim. Note that a standard HRTEM cannot be used to resolve the top atomic layer of nanoparticles. Second, an amorphous layer does not have to be porous, for example, amorphous silicon. Third, why does 6H-SrIrO₃ show a negative cm³/g in Figure S17?

Comment 6c: The authors' response is not satisfactory. There are several concerns. First, there are many types of amorphous IrO_x. Showing that the CV of 6H-SrIrO₃ does not match one does not mean that the surface is not made of other types of amorphous IrO_x. Second, both the CVs of 6H-SrIrO₃ and 3C-SrIrO₃ change significantly after the OER. This indicates surface restructuring and is a sign of amorphization. If anything, the CVs actually suggest that the surface of 6H-SrIrO₃ is amorphous. Third, the authors claim that "the electrochemical active surface area of 6H-SrIrO₃ is about 3 times lower than that of IrO₂". However, Figure S18 shows that the ECSA's are comparable after the OER. Please show the calculation.

Comment 7 and 9: The responses to these comments are tied to whether the surface of 6H-SrIrO₃ is

amorphous or not. If the authors can conclusively show that the surface of 6H-SrIrO₃ is not amorphous, the responses are satisfactory. If not, the claim should be revisited in the revision.

Reviewer #2's Comments and Our Responses:

Reviewer: 2

Comment: The authors have not presented sufficient evidence to conclusively show that the surface of 6H-SrIrO₃ is not amorphous. If the authors can provide the evidence to support this claim, most of the concerns will be addressed. The authors have so far presented HRTEM, which cannot be used to resolve the topmost atomic layer of nanomaterials, and CV, which show that the surface chemistry changes before and after the OER. The authors argued based on the comparison between the CV of 6H-SrIrO₃ and IrO_x that the surface of 6H-SrIrO₃ is not amorphous. However, as there are many types of amorphous IrO_x (which is why some amorphous IrO_x are more active than the others, for example, in Ref. 16), one comparison is not sufficient and cannot be used to conclude that the surface is crystalline. For this reason and other weaknesses in the lack of information on the synthesis side, I cannot recommend an acceptance of the work until all the concerns are fully and entirely addressed.

Response: We appreciate the further comments of Reviewer #2, which have enabled us to improve the manuscript. We have addressed the comments point-by-point as follows.

In order to address the main concern of the reviewer, we studied the 6H-SrIrO₃ before and after OER using high-angle annular dark field scanning transmission electron microscopy. The results are provided in Figures S7 and S20 in Supporting Information. The results suggest that there is not secondary, amorphous IrO_x phase generated on the 6H-SrIrO₃ surface before and after OER.

Some important discussion and characterization details are added in the manuscript and SI (or see below).

“...High-angle annular dark field scanning transmission electron microscopy (STEM) images were acquired using an aberration-corrected STEM with the model of FEI Titan Cubed Themis G2 300, whose accelerating voltage and electron current were set at 200 kV and around 40 pA, respectively. (*Note that because 6H-SrIrO₃ is very sensitive to the electron beams, high-angle annular dark field STEM images must be obtained at the lowest electron current of 40 pA.*)” (see Page 3 of SI)

“...Furthermore, high-angle annular dark field (HAADF) STEM image (Figure S7 in SI) of the edge of a 6H-SrIrO₃ particle reveals that the 6H-SrIrO₃ is highly crystalline without amorphous layers on the surface.” (see Page 7 of manuscript)

“...This is further supported by the high-angle annular dark field (HAADF) STEM image (Figure S20, SI), which shows that there is not secondary, amorphous IrO_x phase generated on the 6H-SrIrO₃ surface after OER.” (see Page 14 of manuscript)

Comment 2: The authors' response is satisfactory. I suggest that the authors include a claim that the sequence of synthesis steps was refined through trials and errors (or other wordings of that nature) in the revision. This will help clearly indicate that there is no specific reason why the

authors took several steps to make 6H-SrIrO₃, which due to its complex series of steps can be confusing to the readers.

Response 2: We have added the relevant claim in the Methods (see below or see Page 19 of manuscript).

“...the sequence of synthesis steps was refined through trials and errors.”

Comment 3: The extra experiments are appreciated. However, the response is not satisfactory. The authors are indirectly suggesting that the presence of a citric acid favors the formation of 3C-SrIrO₃ by stabilizing the more oxidized form of iridates. First, the authors should directly state this claim instead of coyly saying “some experiments were performed”. Second, the citric acid is a reducing agent. Should not it drive iridium toward a more reduced state, which would consequently destabilize 3C-SrIrO₃?

Response 3: First, we thank for the reviewer’s suggestion. We directly state our claim in the manuscript (see Page 12 of manuscript or see below).

“...Overall, the presence of more citric acid favors the formation of 3C-SrIrO₃ by facilitating the generation of the more oxidized iridium species.”

Second, in order to answer the question of the reviewer, we first prepared two samples by calcining the precursors of 6H-SrIrO₃ and 3C-SrIrO₃ at 500 °C. The selection of this temperature for the sample preparation is because: at 500 °C citric acid in the precursors is oxidatively decomposed in air (see Figure S13a, SI), and calcination at 500 °C is the important step before the formation of 6H-SrIrO₃ and 3C-SrIrO₃ (see details in Methods). After getting the samples, we then determined the oxidation state of iridium in them by XPS. The results (Figure S13b, SI) show that the sample obtained by calcining the precursor of 6H-SrIrO₃ at 500 °C contains zero-valent and four-valent Ir species, whereas the sample obtained by calcining the precursor of 3C-SrIrO₃ at 500 °C contains four-valent and five-valent Ir species. These results indicate that when a small amount of citric acid was used for the synthesis, a reduction reaction dominates the oxidation state of iridium, whereas when a large amount of citric acid was used for the synthesis, such a reduction reaction is completely inhibited. The presence of a large amount of citric acid (*i.e.*, the precursor for the synthesis of 3C-SrIrO₃) is found to result in more heat released at 500 °C due to the combustion reaction (Figure S13a, SI). The more released heat in the synthesis should favor the oxidation of iridium in air, generating high-valent iridium species before the formation of 3C-SrIrO₃.

Comment 4: The response is not satisfactory. It is a common understanding that smaller particles have higher surface areas. Stating the obvious fact that materials with higher surface areas have smaller sizes is not helpful and does not address the original question of why the areas/sizes are different.

Response 4: In order to answer the reviewer’s question, we additionally prepared two samples by calcining the precursor of IrO₂ at 300 and 500 °C, respectively. Their XRD patterns are provided in Figure S27 in SI. The results show that IrO₂ starts crystallization at a very low temperature of 300 °C, and with the increase in the reaction temperature, the degree of crystallinity of IrO₂ increase. By comparing the synthesis of IrO₂ and 6H-SrIrO₃, we can find that although their experimental procedures are almost the same in our work, the crystallization processes are totally different for IrO₂ and 6H-SrIrO₃. IrO₂ can crystallize at a very low temperature of 300 °C, but 6H-SrIrO₃ crystallizes until the temperature increase up to 700 °C. This might be the reason why the experimental procedures are almost the same for the synthesis of IrO₂ and 6H-SrIrO₃ in our work, but their areas/sizes are different.

Figure 1. N₂ adsorption–desorption isotherms of 6H-SrIrO₃ after OER, with different amount of samples for testing.

Comment 6b: The authors’ response is not satisfactory. There are several concerns. First, the authors claim that there is no amorphous IrO_x layer. Please provide experimental evidence to support this claim. Note that a standard HRTEM cannot be used to resolve the top atomic layer of nanoparticles. Second, an amorphous layer does not have to be porous, for example, amorphous silicon. Third, why does 6H-SrIrO₃ show a negative cm³/g in Figure S17?

Response 6b: First, we studied the 6H-SrIrO₃ before and after OER using high-angle annular dark field scanning transmission electron microscopy. The results can support our claim: there is not amorphous layer on the surface of 6H-SrIrO₃.

Second, I agree with the reviewer's concern 2. Your comment is quite right.

Third, the negative cm³/g is originated from the instrumental errors. This phenomenon is usually observed for the material with a very low surface area, like 6H-SrIrO₃. In order to minimize instrumental errors, we increased the sample amount for the testing. As shown in the attached Figure 1, we can get satisfied N₂ adsorption–desorption isotherms when the sample amount for testing increases to 300 mg. Correspondingly, we have revised the N₂ adsorption–desorption isotherms in the Figure S18 in SI.

Comment 6c: The authors' response is not satisfactory. There are several concerns. First, there are many types of amorphous IrO_x. Showing that the CV of 6H-SrIrO₃ does not match one does not mean that the surface is not made of other types of amorphous IrO_x. Second, both the CVs of 6H-SrIrO₃ and 3C-SrIrO₃ change significantly after the OER. This indicates surface restructuring and is a sign of amorphization. If anything, the CVs actually suggest that the surface of 6H-SrIrO₃ is amorphous. Third, the authors claim that “the electrochemical active surface area of 6H-SrIrO₃ is about 3 times lower than that of IrO₂”. However, Figure S18 shows that the ECSAs are comparable after the OER. Please show the calculation.

Response 6c: First, we agree with the reviewer's comment on the amorphous IrO_x. The amorphous IrO_x is indeed not a suitable reference material. We removed the CV curves of the amorphous IrO_x from Figure S19, SI.

Second, we revised our discussion on the CV curves (please see Page 26 of SI and Page 14 of manuscript or see below).

“As shown in Figure S19, the CV curves are not complete overlap before and after OER for all the three samples. Different from that of 3C-SrIrO₃, the CV shapes of 6H-SrIrO₃ and IrO₂ does not change obviously after OER, indicating that the later two materials do not undergo significant variation in surface structure. The slight increase in the area of CV curve for 6H-SrIrO₃ after OER might be due to the slight surface Sr leaching during OER.” (Page 26 of SI)

“Moreover, the comparison of the 6H-SrIrO₃'s CV curves with IrO₂ and 3C-SrIrO₃ before and after OER (Figure S19, SI) suggests that different from 3C-SrIrO₃, 6H-SrIrO₃ does not undergo significant variation in surface structure because the CV shape of 6H-SrIrO₃ does not change obviously after OER. This is further supported by the high-angle annular dark field (HAADF) STEM image (Figure S20, SI), which shows that there is not secondary, amorphous IrO_x phase generated on the 6H-SrIrO₃ surface after OER.” (Page 14 of manuscript)

Third, the electrochemically active surface area (ECSA) of each sample was obtained by determining the double-layer capacitance at non-Faradaic potential range, according to the method reported by Jaramillo *et al.* (see reference: Benchmarking hydrogen evolving reaction

and oxygen evolving reaction electrocatalysts for solar water splitting devices. *J. Am. Chem. Soc.* 2015, **137**, 4347–4357.) The experimental details are added in the Methods (see Page 21 of manuscript). Additionally, the original experimental data are added in the Figure S10, SI.

Comment 7 and 9: The responses to these comments are tied to whether the surface of 6H-SrIrO₃ is amorphous or not. If the authors can conclusively show that the surface of 6H-SrIrO₃ is not amorphous, the responses are satisfactory. If not, the claim should be revisited in the revision.

Response 7 and 9: Thanks for the reviewer's comments. We think that our high-angle annular dark field STEM results can clearly show that the surface of 6H-SrIrO₃ is not amorphous.

We hope that we have addressed the concerns raised by the reviewers. We look forward to hearing your decision at your convenience please. Thanks a lot!

With best regards,

Sincerely yours,

Xiaoxin Zou

REVIEWERS' COMMENTS:

Reviewer #2 (Remarks to the Author):

The authors have satisfactorily addressed most of the concerns. The remaining major concern is that it is unclear if the HAADF measurement is sufficient to show that the surface of 6H-SrIrO₃ is not amorphous. Since the amorphous IrO_x layer is likely very thin and the Ir atoms in the amorphous structure do not line up along the crystallographic direction, their HAADF signals should be very weak. I recognize that this is a hard question. So, my suggestion instead is to ask the authors to add a paragraph that the possibility of an amorphous surface IrO_x layer cannot be ruled out. The result on this work is valuable to the community, so it is not necessary that the authors understand everything. This is also the situation with Ref 16. However, they authors should not make claims based on ambiguous results. An acceptance is recommended provided that the authors address this concern and the points below.

General Remark: I do not think that HAADF can be used to resolve amorphous vs. crystalline material. The authors may see only the crystalline component, since the amorphous IrO_x signal may be relatively weaker due to the lack of the zone axis and is thin, so small in the number of atoms. One way to resolve this question is to show that HAADF can resolve amorphous IrO_x by doing HAADF on IrO_x and a mixture of IrO_x and 6H-SrIrO₃ (mechanically mixed). If the authors can show that HAADF can resolve individual Ir atoms in amorphous IrO_x such that they can be differentiated from 6H-SrIrO₃, this concern is addressed.

Comment 4: The authors should do a similar temperature-dependent XRD experiment for 6H-SrIrO₃ for comparison with IrO₂. In Figure S27, there should be a side-by-side comparison of XRD of IrO₂ vs. 6H-SrIrO₃ at different temperature.

Comment 6c: On the second point, it is misleading to claim that the CV of 6H-SrIrO₃ does not change after OER. This is simply not true. The double layer of IrO_x generally shows up at low potential (0 – 0.4 V), while the redox typically shows up at high potential (> 0.4 V). Figure S19 shows that the double layer capacitance does not change but the Ir redox does.

Our Responses to the referees' comments:

Reviewer: 2

Comment 1: The authors have satisfactorily addressed most of the concerns. The remaining major concern is that it is unclear if the HAADF measurement is sufficient to show that the surface of 6H-SrIrO₃ is not amorphous. Since the amorphous IrO_x layer is likely very thin and the Ir atoms in the amorphous structure do not line up along the crystallographic direction, their HAADF signals should be very weak. I recognize that this is a hard question. So, my suggestion instead is to ask the authors to add a paragraph that the possibility of an amorphous surface IrO_x layer cannot be ruled out. The result on this work is valuable to the community, so it is not necessary that the authors understand everything. This is also the situation with Ref 16. However, they authors should not make claims based on ambiguous results. An acceptance is recommended provided that the authors address this concern and the points below.

General Remark: I do not think that HAADF can be used to resolve amorphous vs. crystalline material. The authors may see only the crystalline component, since the amorphous IrO_x signal may be relatively weaker due to the lack of the zone axis and is thin, so small in the number of atoms. One way to resolve this question is to show that HAADF can resolve amorphous IrO_x by doing HAADF on IrO_x and a mixture of IrO_x and 6H-SrIrO₃ (mechanically mixed). If the authors can show that HAADF can resolve individual Ir atoms in amorphous IrO_x such that they can be differentiated from 6H-SrIrO₃, this concern is addressed.

Response 1: Thanks for the reviewer's valuable comments. We agree with the comments on the necessity of discussing the possibility of an amorphous surface IrO_x layer. Thus, we add the following discussion in the revised manuscript (see Page 15).

"...It should be pointed out that while the better structural stability of 6H-SrIrO₃ than 3C-SrIrO₃ has been unambiguously confirmed, the possibility of a very thin amorphous IrO_x layer (or a tiny amount of IrO_x clusters) on the 6H-SrIrO₃ surface cannot be completely ruled out at current stage."

Comment 4: The authors should do a similar temperature-dependent XRD experiment for 6H-SrIrO₃ for comparison with IrO₂. In Figure S27, there should be a side-by-side comparison of XRD of IrO₂ vs. 6H-SrIrO₃ at different temperature.

Response 4: We add the temperature-dependent XRD results for 6H-SrIrO₃ in Supplementary Figure 27.

Comment 6c: On the second point, it is misleading to claim that the CV of 6H-SrIrO₃ does not change after OER. This is simply not true. The double layer of IrO_x generally shows up at low

potential (0-0.4 V), while the redox typically shows up at high potential (> 0.4 V). Figure S19 shows that the double layer capacitance does not change but the Ir redox does.

Response 6c: Thanks! We modified the discussion on Supplementary Figure 19 (please see Page 14 or see below).

“...the comparison of the 6H-SrIrO₃'s CV curves before and after OER (Supplementary Figure 19) shows that the CV shape of 6H-SrIrO₃ after OER is not exactly same as that before OER, but does not change a lot. This suggests that 6H-SrIrO₃ might undergo a weak surface reconstruction during OER probably thanks to the slight Sr leaching during OER...”